# Snow depth in high-resolution regional climate model simulations over southern Germany - suitable for extremes and impact-related research?

Benjamin Poschlod[1], Anne Sophie Daloz[2]

[1]Research Unit Sustainability and Climate Risk, Center for Earth System Research and Sustainability (CEN), Universität Hamburg, 20144 Hamburg, Germany
[2]Center for International Climate Research (CICERO), 0349 Oslo, Norway

*Correspondence to*: Benjamin Poschlod (benjamin.poschlod@uni-hamburg.de)

**Abstract.** Snow dynamics play a critical role in the climate system as they affect the water cycle, ecosystems and society. Within climate modelling, the representation of the amount and extent of snow on the land surface is crucial for simulating the mass and energy balance of the climate system. Here, we evaluate simulations of daily snow depths against 83 station observations in southern Germany in an elevation range of 150 m to 1000 m over the time period 1987 – 2018. Two simulations stem from high-resolution regional climate models, the Weather Research & Forecasting Model (WRF) at 1.5 km resolution and the COSMO-CLM (CCLM) at 3 km resolution. Additionally, the hydrometeorological snow model AMUNDSEN is run at the point scale of the climate stations based on the atmospheric output of CCLM. The ERA5-Land dataset (9 km) complements the comparison as state-of-the-art reanalysis land surface product. All four simulations are driven by the same atmospheric boundary conditions of ERA5.

Due to an overestimation of snow albedo, the WRF simulation features a cold bias of 1.2°C leading to slight overestimation of snow depth in low-lying areas, whereas the snow depth is underestimated at snow-rich stations. The number of snow days (days with snow depth above 1 cm) is well reproduced. The WRF simulation can recreate extreme snow depths, i.e., annual maxima of snow depth, their timing and inter-station differences, thereby showing the best performance of all models.

The CCLM reproduces the climatic conditions with very low bias and error metrics. However, all snow-related assessments show a strong systematic underestimation, which we relate to deficiencies in the snow module of the land surface model. When driving AMUNDSEN with the atmospheric output of the CCLM, the results turn to a slight tendency of overestimation for snow depth and number of snow days, especially in the northern parts of the study area. Snow depth extremes are well reproduced.

For ERA5-Land (ERA5L), the coarser spatial resolution leads to larger differences between model elevation and station elevation, which contributes to climatic biases significantly correlating with the elevation bias. In addition, the mean snow depth and number of snow days are strongly overestimated, with too snowy conditions in the late winter. Extreme snow depth conditions are well reproduced in the low-lying areas, whereas strong deviations occur in more complex topography.

In sum, the high spatial resolution of convection-permitting climate models shows potential in reproducing the winter climate (temperature and precipitation) in southern Germany. However, different sources of uncertainties, i.e. spatial resolution, snow albedo parametrisation, and other parametrisations within the snow modelling prevent a further straightforward use for impact research. Hence, careful evaluation is needed before any impact-related interpretation of the simulations, also in the context of climate change research.

## 1 Introduction

The presence and absence of snow, as well as snow depth, affect nature and humans on many different levels. The albedo of snow influences the radiation and energy balance of the surface (Warren, 2019). Its insulating effect protects plants and animals (Blume-Werry et al., 2016; Slatyer et al., 2022). In addition, snow cover affects the microbial structure of the soil (Gavazov et al., 2017). The seasonal cycle of snow cover duration and snow depth governs soil moisture dynamics (Qi et al., 2020) and also the runoff regime of rivers and streams (Girons Lopez et al., 2020; Poschlod et al., 2020a). Hence, the snow dynamics also affect the freshwater availability in large regions of the world (Barnett et al., 2005). Snow melt may also induce riverine floods (Berghuijs et al., 2019), often in combination with heavy rainfall (Poschlod et al., 2020b). Sufficient snow depths are needed for ski tourism (Steiger et al., 2017; Witting and Schmude, 2019), but also expected by non-ski tourists during their winter holidays in the mountains (Bausch and Unseld, 2018). This reflects also a traditional and cultural meaning of snow cover, which manifests annually in discussions and expectations about "white Christmas" in the Northern Hemisphere (Durre and Squires, 2015; Harley, 2003). Moreover, the presence of snow influences everyday life in terms of mobility. Usage of bikes or scooters is reported to significantly depend on snow depths (Mathew et al., 2019; Yang et al., 2018). Also, the alternatives (trains, cars, planes) may be affected by high snow depths (Doll et al., 2014; Taszarek et al., 2020; Trinks et al., 2012). During extreme snow events, power shortages (Bednorz, 2013; Gerhold et al., 2019) or even collapses of roofs (Strasser, 2008a) occur, which is why building codes have to rely on the regional snow climate (Croce et al., 2018).

Due to these manifold effects, there is great interest in modelling snow dynamics and snow depths in order to be able to predict near-term snow conditions (Hammer, 2018) and to project future snow conditions (Frei et al., 2018). Snow models in general are used to simulate snow dynamics on different temporal and spatial scales. In global or regional earth system model setups, snow dynamics are simulated coupled with atmospheric processes and other land surface processes (Krinner et al., 2018). However, the spatial resolution of global and regional models is often not sufficient to represent complex terrain and the spatial variability of the land surface (Mooney et al., 2022). Further, the simulated climate and snow dynamics show biases (Daloz et al. 2022). Often, for local to regional impact studies, the climatic biases are adjusted, where it is necessary to apply a method tailored to snow climates (e.g. Chen et al., 2018; Frei et al., 2018; Meyer et al., 2019). In the case of a simple univariate bias adjustment, the dependence between temperature and precipitation is not considered, which neglects the threshold effect of air temperature on fractionation of precipitation into rain and snowfall (Meyer et al., 2019). Therefore, multivariate bias adjustment is recommended for hydrological impact modelling in regions with snow and rain dynamics (Chen et al., 2018;

Meyer et al., 2019). In a next step, snow models are set up at higher resolution, driven by bias-adjusted climatic time series (Hanzer et al., 2018). However, then the simulated snow dynamics cannot feed back into the climate, which is why these simulations are called "offline". Another possibility would be to directly adjust the biases of the snow parameter simulated by the climate model (Matiu and Hanzer, 2022).

Due to the advances in computational power, the spatial resolution of regional climate models (RCMs) has increased (Coppola et al., 2020). Kilometre-scale simulations are available for decade-long time spans. The high spatial resolution allows for a finer-scale representation of complex topography (Poschlod et al., 2018). Therefore, the effects of altitude on air temperature can be mapped in more detail, which better represents the fractionation into rain and snowfall, as well as melting and accumulation processes. In addition, the higher resolution of land cover allows for more detailed simulation of the albedo, which in turn governs the energy balance (Winter et al., 2017). High resolution also allows to include more processes in the model chain. Snow drift due to wind and turbulence has been implemented in an offline setup (Vionnet et al., 2021) and also online by coupling the RCM Weather Research and Forecasting (WRF) model with the detailed snow model SNOWPACK (Sharma et al., 2023). Future work aims at seamlessly implementing snow drift in WRF (Saigger et al., 2023).

Here, two high-resolution convection-permitting regional climate models are evaluated with station observations of daily snow depth in southern Germany for a 31-year period within 1987 – 2018. We analyse simulations of the WRF model (Skamarock et al., 2019) at 1.5 km resolution and the COnsortium for Small scale MOdelling (COSMO; Sørland et al., 2021) model in climate mode (CLM) at 3 km resolution (hereafter abbreviated to CCLM). Both models are driven by atmospheric boundary conditions of ERA5 (Hersbach et al., 2020) at 31 km resolution. Moreover, the hydrometeorological snow model AMUNDSEN is run at the point scale of the climate stations driven by the atmospheric output of CCLM. In addition, the state-of-the-art land surface reanalysis product ERA5-Land (9 km resolution; Muñoz-Sabater et al., 2021) is compared.

Such evaluation is important as future projections of snow dynamics are often based on regional climate models (e.g. Frei et al., 2018; Räisänen, 2021). Even if mean snow depth and mean snow cover duration are expected to decrease due to higher temperatures, risks associated with snow dynamics might not (Musselman et al., 2018). While there are studies on climate model simulations and extreme snowfall (Quante et al., 2021; Sasai et al., 2019), we find no literature about extremes of snow depth dynamics within climate model simulations. Furthermore, there are studies evaluating regional climate models based on multi-decadal simulations, however at coarser spatial resolution (Daloz et al., 2022: 0.11°; Matiu and Hanzer, 2022: 0.11°; Steger et al., 2013: 0.22°). Recently, Monteiro and Morin (2023) compare the performance of multiple model systems in a range of 2.5 to 30 km spatial resolution over the European Alps. They find that main features of snow cover, snow depth, and driving climatic conditions can be reproduced by the models, however with increasing deviations at higher altitude. Lüthi et al. (2019) compared a 10-year simulation of COSMO at 2.2 km over Switzerland to interpolated observations. There, the mean seasonal cycle averaged over the country is well reproduced, where the model overestimates mean snow water equivalents (SWE) at high altitudes.

In contrast to these existing high-resolution studies in Alpine terrain, we aim to assess snow conditions in southern Germany, mainly north of the Alpine crest at a lower elevation range between 150 m to 1000 m. Even though topography is less complex,

snow dynamics still play a major role in this area affecting natural systems (Poschlod et al., 2020a) and human systems (Strasser, 2008a; Doll et al., 2014; Frese and Blaß, 2011). Snow plays an important role for the tourism industry, where the sufficient presence of snow is not only important for ski tourists at higher altitude, but any winter tourists (Bausch and Unseld, 2018; Witting et al., 2021). Due to the population density, there is a high exposure to potential snow extremes. Within Germany, the study area covers a variety of snow load zones, which are used as the basis for the structural dimensioning of roofs (German Industry Norm DIN 1055-5; DIN, 2005). In the winter season 2005/2006, continuous snow cover conditions induced several roof collapses in the study area (Strasser, 2008a). During the recent winter (December 2023) snow depths above 40 cm in the study area (HND, 2024) led to a collapse in local and long-distance transport, power shortages and damage to buildings and cars (Hagen and Mese, 2023; ARD, 2023).

Climate change already affects and will further alter snow dynamics and conditions (Dong and Menzel, 2020; Monteiro and Morin, 2023), which is why observation-based analyses are limited. Climate impact research and data users benefit from information at the local scale (Orr et al., 2021). In order to provide local information, often coarse-resolution RCMs or even global circulation models have been used to drive snow models at the local scale, however involving bias adjustment, statistical downscaling and the de-coupling of the interactions of snow dynamics and climate, which induces additional uncertainties and limitations.

The "new generation" of high-resolution RCMs can potentially directly provide snow depth information based on their internal land surface and snow modules. Hence, we see the need for a critical examination of what new-generation high-resolution RCMs are capable of in terms of snow dynamics and extremes. So, in addition to the evaluation of winter temperature and precipitation, mean winter snow depth and duration, which is also present in the above-mentioned studies, the study further aims to explore the capabilities of the high-resolution models to reproduce short-period snow conditions and extreme daily snow depth. The sample size required for this motivates the research setup, where we select available multi-decadal high-resolution simulations (WRF, CCLM, AMUNDSEN driven by CCLM) in comparison to the coarser-resolved product ERA5L. So, in sum the study aims to evaluate 1) winter climate, 2) mean seasonal snow conditions, 3) short-duration snow conditions with relevance for the tourism sector, and 4) extreme snow depths in order to 5) explore the suitability of high-resolution climate models for impact-relevant snow research.

## 2 Data and methodology

A comparative overview of the investigated simulation data is given in Table 1. The following sections describe the different model features and setups in detail. As a preliminary remark, we would like to point out that the historical genesis of the different model setups still governs the degree of complexity of the snow schemes. Single-layer snow schemes are common in the atmospheric community for numerical weather prediction (NWP) models and reanalyses (Arduini et al., 2019). Lee et al. (2023) give an overview of various snow parametrisations within nine land-surface models.

**Table 1: Overview of the different investigated model setups.**

| Setup name | Boundary conditions | Downscaling of the climatic input variables | Spatial resolution | Land surface / snow model | maximal number of snow layers | Range of snow albedo |
|---|---|---|---|---|---|---|
| WRF | ERA5[a] | Dynamical: WRF[b] | 1.5 km | NOAH_MP[c] | 3 | exposed, non-melting: 0.7 to 0.84; exposed, melting: 0.5 to 0.84; forested[d] |
| CCLM | ERA5[a] | Dynamical: COSMO-CLM[e] | 3 km | TERRA-ML[e] | 1 | exposed: 0.4 to 0.7; forested: empirical reduction factor |
| ERA5L | ERA5[a] | Statistical: linear interpolation for ERA5-Land[f] | 9 km | CHTESSEL[g] | 1 | exposed: 0.5 to 0.85; forested: 0.27 to 0.38 |
| AMUNDSEN | ERA5[a] | Dynamical: COSMO-CLM[e] | 3 km | AMUNDSEN[h,i] | 3 | exposed: 0.55 to 0.85 |

[a]Hersbach et al. (2020), [b]Skamarock et al. (2019), [c]Niu et al. (2011), [d]Niu and Yang (2004), [e]Doms et al. (2021), [f]Muñoz-Sabater et al. (2021), [g]ECMWF (2018) [h]Strasser (2008b) [i]Hanzer et al. (2018)

## 2.1 Regional climate models

Both RCMs are driven by atmospheric boundary conditions of the ERA5 reanalysis at 31 km resolution. The WRF simulations have been carried out by Collier and Mölg (2020), where the setup is described in detail. The WRF is run in version 4.1

(Skamarock et al., 2019). The boundary conditions by ERA5 drive the WRF at $7.5 \times 7.5$ km² (one-way nesting), where spectral nudging is applied. The forcing at the lateral boundaries is updated at 3-hourly steps. Increasing the spatial resolution by factor five within a second nesting step results in simulations at $1.5 \times 1.5$ km² resolution, where convective processes are explicitly resolved. Collier and Mölg (2020) provide a general climatic evaluation with observational data. In order to simulate surface processes WRF is run coupled with the land surface scheme NOAH_MP (Niu et al., 2011). The physically-based snow model

in NOAH_MP features up to three snow layers, which are divided based on the simulated snow depth. In addition, four soil levels and a vegetation canopy layer are considered. Interception and burial of vegetation by snow is modelled following Niu and Yang (2004). Density of newly fallen snow is modelled dependent on the atmospheric temperature (Niu et al., 2011). Based on the energy balance for the snow and soil layers, the temperatures of the layers are calculated. For above/below 0°C, melting and freezing are assumed. Mass and energy transfer between layers are accounted for (Niu et al., 2011). In order to

derive the snow depth, the density is assessed representing snow metamorphism and compaction following Anderson (1976)

and Sun et al. (1999). The snow surface albedo is calculated following the CLASS scheme (Verseghy, 1991), where the albedo of fresh snow is assumed to amount to 0.84. During melting conditions, the albedo exponentially decreases with 0.5 as lower limit. Without melting conditions, 0.7 is assumed as lower limit. The snow albedo of forested areas is modelled dependent on the leaf and stem area index accounting for snow interception, loading and unloading of snow as well as melting and refreezing

(Niu and Yang, 2004). The ground surface albedo is calculated as an area-weighted average of the snow albedo and bare soil albedo (Niu et al., 2011). The WRF data are openly available (Collier, 2020).

COSMO was operationally applied as weather forecasting model in Germany over 20 years being replaced by ICON (Rybka et al., 2022). For regional climate simulations, the ICON-CLM is developed (Pham et al., 2021). Here, still the CCLM version 5-0-16 at 3 km resolution (Brienen et al., 2022) is directly driven by the atmospheric conditions from ERA5, where the forcing

is updated every hour (Rybka et al., 2022). The simulation domain covers Germany, surrounding catchments and parts of the Alps. Hence, the analysis domain is well within the simulation domain. The 3 km resolution allows for explicit simulation of deep convection, whereby shallow convection is parametrised. The land surface model TERRA_ML is implemented in COSMO-CLM and represents soil, vegetation and snow dynamics (Doms et al., 2021, Schulz and Vogel, 2020). It features multiple soil layers and either one interception reservoir or one snow reservoir on top of the soil, depending on the predicted

temperature of the stored water. However, no canopy layer is implemented, which is common in NWP models, but represents a strong simplification for climate models (Schulz et al., 2016). Mass and energy fluxes between the atmosphere, the interception/snow reservoir and the soil layers govern the temperature of the water in the reservoir. In case of snow cover, the mean snow temperature is calculated based on the heat capacity of the snow, the atmospheric forcing at the snow surface, the heat flux to the soil, and melting processes (Doms et al., 2021). The density of fresh snow is assumed in the range of 50 and

150 kg/m³ dependent on temperature conditions. The empirical estimation of the mean snow density within the single snow layer includes two processes. Aging dependent on snow temperature and time increases the mean snow density of the reservoir, whereas newly fallen snow decreases the mean density. The range of densities is fixed between 50 and 400 kg/m³. Doms et al. (2021) claim that extreme snow depths cannot be properly accounted for by the model concept. The range of possible snow depths is restricted between 0.01 to 1.5 m. A time-dependent snow surface albedo is calculated, where the albedo covers the

range between 0.4 for old snow and 0.7 for fresh snow (Doms et al., 2021). As the model features no canopy layer, no shading effects of vegetation are simulated. Instead an empirical reduction factor for the snow albedo under vegetation is applied (Daloz et al., 2022). The CCLM data are available publicly (see https://esgf.dwd.de/projects/dwd-cps/). However, albedo data were provided separately without quality checks (pers. comm. Susanne Brienen, German Weather Service).

**2.2 Land surface model in ERA5-Land**

ERA5-Land (ERA5L) directly derives its atmospheric forcing from the 10 m level of ERA5 at hourly resolution (Muñoz-Sabater et al., 2021). The 31 km resolution of ERA5 is linearly interpolated on the 9 km grid of ERA5-Land. The land surface model Carbon Hydrology-Tiled ECMWF Scheme for Surface Exchanges over Land (CHTESSEL) then simulates energy and water cycles globally over land at hourly resolution. The snow scheme features one layer with a single temperature and density

on top of the four-layer soil (ECMWF, 2018). The snow temperature is calculated based on the energy fluxes between the

atmosphere and the snow skin, melting processes, and basal heat flux. Snow temperature, mass, and density are used to derive the liquid water content within the snow pack. Interception of liquid water by the snow pack is accounted for. Snow density of fresh snow is modelled dependent on atmospheric temperature and near-surface wind speed (Dutra et al., 2011). The snow density is assumed to change due to overburden, thermal metamorphisms (Anderson, 1976) and compaction (Lynch-Stieglitz, 1994). The range of snow densities is restricted between 50 and 450 kg/m³. The calculation of the snow surface albedo for

exposed areas follows Verseghy (1991) with values between 0.5 and 0.85. For forested areas, lower albedo values between 0.27 and 0.38 are applied following Moody et al. (2007) who based their calculations on remote sensing products from MODIS. Further details of the land surface scheme are provided by ECMWF (2018) and especially for the snow scheme by Dutra et al. (2010). The ERA5L data are openly available (https://cds.climate.copernicus.eu/cdsapp#!/dataset/reanalysis-era5-land?tab=overview). For the next land surface model generation, Bousetta et al. (2021) present the new ECLand model, which

introduces a multi-layer snow scheme following Arduini et al. (2019).

## 2.3 Hydroclimatological model AMUNDSEN

The hydroclimatological model AMUNDSEN has been developed to dynamically resolve the mass and energy balance of snow and ice in high-mountain regions (Strasser, 2008b; Hanzer et al., 2018). The model can be set up spatially distributed at high resolution (e.g. 10 m to 100 m) and includes snow redistribution processes and a radiation scheme accounting for e.g.

terrain slope and hill shading (Hanzer et al., 2016). Here, we apply the model at point scale at the locations of the 83 climate stations. We use the CCLM output of hourly temperature, precipitation, wind speed, relative humidity and shortwave radiation to drive openAMUNDSEN, the open model version implemented in Python (Warscher et al., 2021). No model-internal correction for precipitation undercatch is applied as the climatological input stems from simulated precipitation by the CCLM. AMUNDSEN uses the wet-bulb temperature to differentiate between solid and liquid precipitation (Hanzer et al., 2018). The

model features three snow layer categories, which are named "new snow", "old snow", and "firn", depending on the snow density and age. The fresh snow density is calculated following Anderson (1976) and Jordan (1991) dependent on air temperature. Compaction and metamorphism also follow empirical formulations by Anderson (1976) and Jordan (1991). Snow with a density above 200 kg/m³ is transferred to the old snow layer. The firn layer does not apply for the 83 locations as no multi-year snow cover is simulated. Melting water in the snowpack may be retained applying a parametrisation by Braun

(1984). Snow albedo is parametrised depending on the age of the snow in a range between 0.55 and 0.85 for exposed surfaces following Hanzer et al. (2016). Snow-free albedo is set to 0.23 representing grassland according to the climate station conditions.

## 2.4 Study area and observational data

The study area covers large parts of southern Germany, where the boundaries are given by the smallest simulation domain of

the WRF model and the national borders. It includes a wide range of elevations between roughly 100 m and 3000 m above sea

level (see Fig. 1a) with therefore different snow dynamics. For the comparison to snow depth observations, we select 83 climate stations in an elevation range between 150 m and 1000 m with less than 30 % missing data, which are operated by the German Weather Service (DWD, 2023). In order to assess the climatology, temperature and precipitation is analysed as well, where precipitation is not corrected for undercatch. The mean annual temperatures of the locations range from 5.4 °C to 10.9 °C and the mean annual precipitation covers the range between 580 mm and 1700 mm. The analysis period spans from November 1987 to April 2018 yielding 31 extended winter seasons. In this study, the extended winter season is defined as six-month period from November to April. All analysis is carried out at daily resolution.

The comparison between measured snow depth at the climate station and the simulated gridded snow depth is carried out via nearest-neighbour approach. As the elevation within the gridded models differs according to the spatial resolution (Fig. 1b-d; see also Fig. S1), the elevation of the climate station and the nearest model grid cell may deviate. Figure 2 shows to what degree the finer spatial resolution improves the representation of the altitude. The mean absolute deviation amounts to 24 m (WRF), 42 m (CCLM), and 93 m (ERA5L).

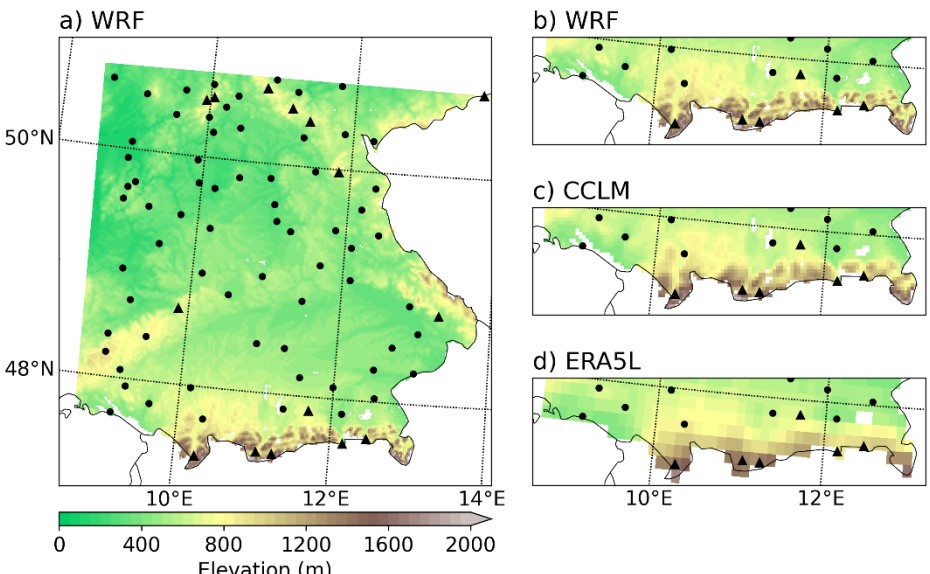

**Figure 1: Representation of elevation in the WRF (a). Lakes are masked out. The black markers show the 83 climate stations, where snow depth is observed. Dots (triangles) refer to stations with less (more) than 5 cm mean snow depth during the extended winter season. Zoomed display of the complex terrain in the southern study area in WRF (b), CCLM (c), and ERA5-Land (d).**

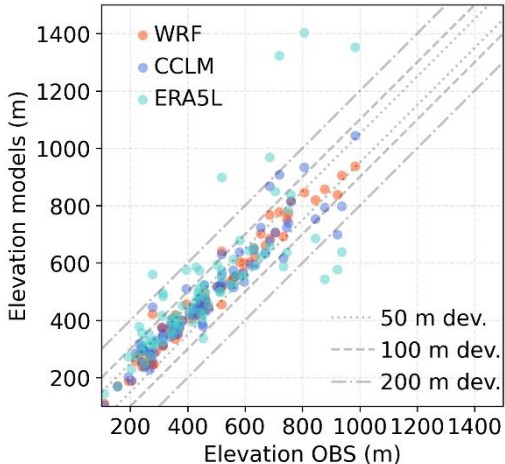

**Figure 2: Comparison of the climate station elevation (OBS) and the model elevation.**

Due to the limited station density, we also include remote sensing data for the model evaluation. Snow depth cannot be directly
derived via optical remote sensing, however the snow cover fraction and the surface albedo. Therefore, we employ the daily
snow cover and albedo from MODIS Terra at 0.05° resolution (Hall and Riggs, 2021; Schaaf and Wang, 2021), which is
available in the period 2000 – 2018. There, we only consider data qualified as "mixed", "okay" or better. Snow depth and
SWE can be derived from microwave remote sensing applying various retrieval algorithms (Tanniru and Ramsankaran, 2023;
Tsang et al., 2022), however with considerable deviations for widely available products (Mortimer et al., 2020). New sensors
and more complex data-driven algorithms have shown the potential to improve these estimations (Daudt et al., 2023; Tsang et
al., 2022), but are out of scope for this evaluation study.

### 2.5 Evaluation criteria

We evaluate the RCMs at the 83 climate stations and calculate the deviations for each winter season. The performance is
assessed with four measures, the mean absolute error (MAE), the root-mean-square error (RMSE), the bias (BIAS) or
percentage bias (PBIAS), and Pearson rank correlation (r). For $n$ observed values $x_{obs}$ and simulated values $x_{sim}$, the measures
are defined as follows:

$$MAE = \frac{\sum_{i=1}^{n} |x_{obs,i} - x_{sim,i}|}{n}, \tag{1}$$

$$RMSE = \sqrt{\frac{\sum_{i=1}^{n} (x_{obs,i} - x_{sim,i})^2}{n}}, \tag{2}$$


$$BIAS = \frac{\sum_{i=1}^{n} x_{sim,i} - x_{obs,i}}{n}, \tag{3}$$

$$PBIAS = \frac{\sum_{i=1}^{n} x_{sim,i} - x_{obs,i}}{\sum_{i=1}^{n} x_{obs,i}} \cdot 100, \tag{4}$$

$$r = \frac{\sum_{i=0}^{n}(x_{obs,i} - \bar{x}_{obs})(x_{sim,i} - \bar{x}_{sim})}{\sqrt{\sum_{i=1}^{n}(x_{obs,i} - \bar{x}_{obs})^2 \sum_{i=1}^{n}(x_{sim,i} - \bar{x}_{sim})^2}}, \tag{5}$$

In addition, we evaluate the simulations regarding intensity and time-related measures of snow dynamics. Thereby, mean winter snow depth refers to the mean snow depth over November to April. We define the number of "snow days" as the number of days with more than 1 cm snow depth considering the whole year.

White Christmas is defined as more than 1 cm snow depth on the 24th, 25th, and 26th of December following the German Weather Service (DWD 2020). As this single 3-day period is rather selective, we also extend the same analysis to a 3-day

moving window between November to April. In addition, we also assess the simulation of 5-day moving windows between December to February, where each of the 5 days is above 10 cm snow depth. This metric reflects the ability to do cross-country skiing (Vassiljev et al., 2010), where 5-day stays are typical for such vacations in the German mountain ranges (Hodeck and Hovemann, 2015). As these criteria lead to a binary classification for each moving day window, Matthews Correlation Coefficient (Matthews, 1975) is used as the evaluation metric. It considers true and false positives (*TP*, *FP*) as well as true and

false negatives (*TN*, *FN*), which is why Luque et al. (2019) recommend it if classification success and errors are to be assessed. The MCC is defined between -1 and 1, where 0 indicates that the classification is as good as a random classifier.

$$MCC = \frac{TP \cdot TN - FP \cdot FN}{\sqrt{(TP+FP)(TP+FN)(TN+FP)(TN+FN)}}, \tag{6}$$

In order to assess extreme conditions, we sample the annual maxima of snow depth. Based on this sampling, we fit the Generalized Extreme Value (GEV) distribution, which can be applied to derive return levels of snow depth, where


$$GEV(x; \xi) = \begin{cases} \exp(-\left[1 + \xi\left(\frac{x-\mu}{\sigma}\right)\right]^{-1/\xi}), \xi \neq 0 \\ \exp(-\exp(-\frac{x-\mu}{\sigma})), \xi = 0, \end{cases} \quad x \in \mathbb{R}, \tag{7}$$

for parameters μ (location), σ (scale) and ξ (shape). The location parameter governs the centre of the distribution, the scale parameter corresponds to its spread, whereas the shape parameter governs the tail behaviour (Coles, 2001). We estimate these GEV parameters using a Markov Chain Monte Carlo algorithm (Bocharov, 2022; Foreman-Mackey et al., 2013) also

generating confidence intervals at the 95 % level.

## 3 Model evaluation

### 3.1 Biases of winter air temperature and precipitation

Precipitation and air temperature govern the snow dynamics to a large degree. The comparison of simulated and observed winter temperature and precipitation is shown in Figure 3. The WRF systematically simulates too low winter temperature,

which has already been noted by Collier and Mölg (2020). Winter temperature is closely reproduced by the CCLM with no bias (Fig. 3b). ERA5L has a slight cold bias, mostly due to single stations, where the underestimation amounts to 4°C (see Fig. 3c) resulting from the elevation bias (Fig. 2). The WRF model slightly underestimates winter precipitation. However, as we do not correct the observed precipitation for undercatch, we would expect a slight overestimation of the RCMs. The CCLM can reproduce winter precipitation with smaller errors, where the positive percentage bias of 9.5 % falls in the range of possible

undercatch-induced deviations in southern Germany (Richter, 1995). The coarser resolved ERA5L shows the biggest deviations with a stronger positive deviation and lowest rank correlation.

The differences of the station elevation and the mean grid cell altitude also contribute to these biases. Elevation governs the temperature and also orographic precipitation effects (Warscher et al., 2019). Hence, Figures 4a-f provide a comparison of the mean 31-year bias per location and the elevation difference. The Pearson rank correlations between elevation (bias) and

temperature or precipitation bias are given in Table 2. For the biases of ERA5L, the relationship is clearly visible (Fig. 4c & 4f), where higher (lower) elevation bias leads to over- (under)estimation of precipitation (r = 0.49) and under- (over)estimation of temperature (r = -0.88). For WRF and CCLM, no strong correlation is found, only a weak correlation of r = 0.22 for the temperature bias in WRF.

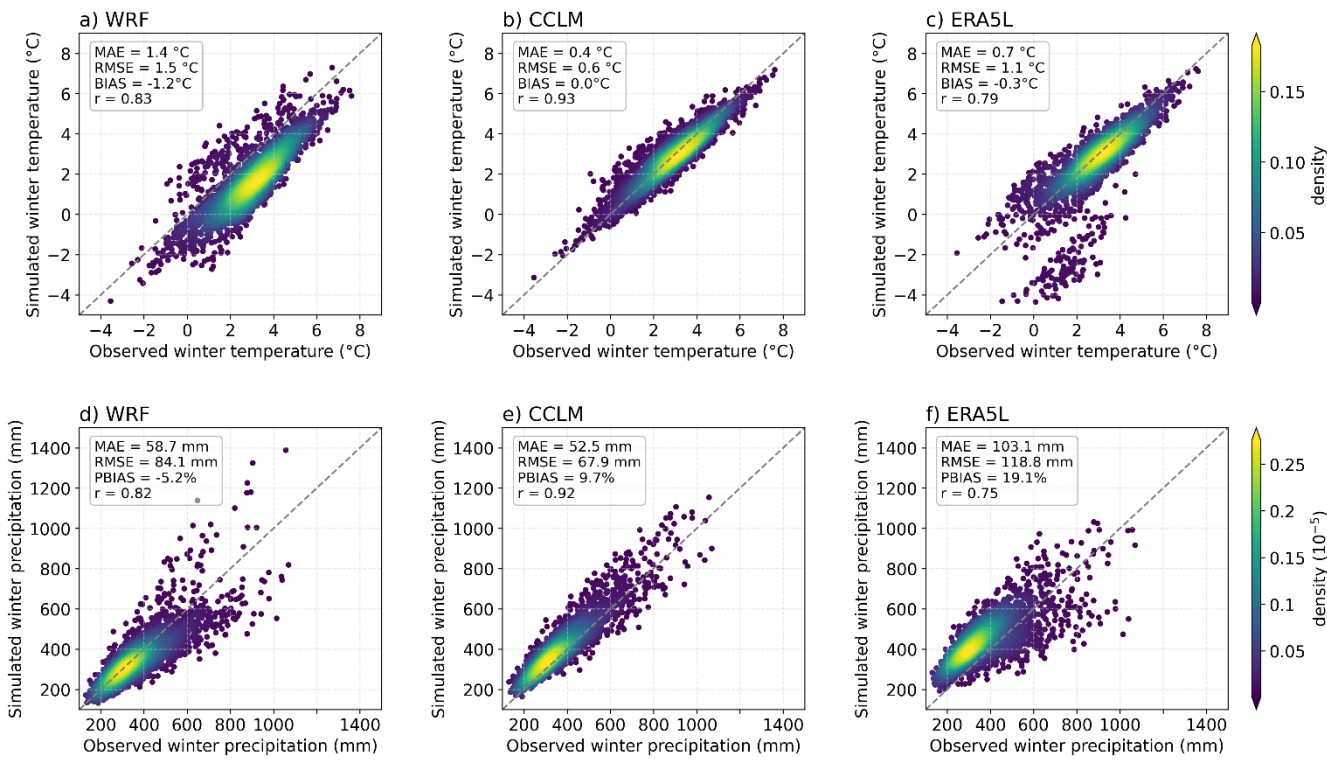


**Figure 3: Mean winter temperature (a-c) and precipitation (d-f) for each extended winter season and location in 1987 – 2018. Simulations by the WRF model (a,d), CCLM (b,e), and ERA5L (c,f) are compared to observations.**

**Table 2: Pearson rank correlation for elevation and climatological biases. Significant correlations at the 95 % (99 %) level are**
**marked with one (two) asterisks.**

| Rank correlation between… | WRF | CCLM | ERA5L |
|---|---|---|---|
| | | | |
| Elevation bias and temperature bias | 0.22* | 0.00 | -0.88** |
| Elevation bias and precipitation bias | -0.03 | -0.18 | 0.49** |
| Model elevation and temperature bias | 0.60** | 0.50** | -0.58** |
| Model elevation and precipitation bias | 0.05 | 0.32** | -0.10 |
| Station elevation and temperature bias | 0.55** | 0.48** | -0.03 |
| Station elevation and precipitation bias | 0.05 | 0.37** | -0.47** |

However, when comparing the absolute model elevation of each location, the CCLM shows systematic deviations (Fig. 4h & 4k). Locations at higher (lower) model elevation tend to have too high (low) temperatures (r = 0.50) and to be too wet (dry; r

= 0.32). The WRF shows higher temperature biases for higher elevations (Fig. 4g) yielding an overall rank correlation of r = 0.60. The negative correlation between ERA5L model elevation and temperature bias (r = -0.58) is governed by the locations at high and medium-high elevation (see Fig. 4i lower right corner), as their temperature bias results from the high elevation bias (see Figs. 2 and 4c).

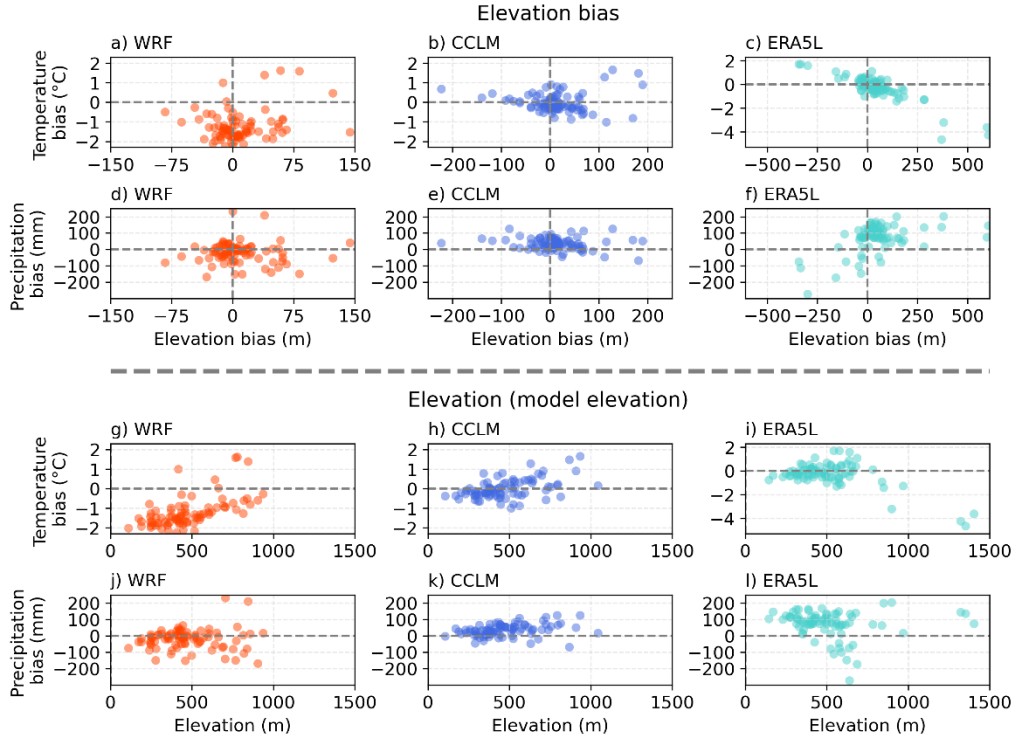

**Figure 4: Relationship of mean winter temperature and precipitation biases against the elevation bias (a-f) and the model elevation (g-l). Each location is averaged over 1987 – 2018. Elevation bias is calculated as model elevation minus station elevation.**

### 3.2 Snow depth evaluation

The evaluation of snow depth time series is challenging due to the high memory of the system and error propagation over time.
Furthermore, a proper evaluation depends on the intended use of the model data. If, for example, snow depth is simulated differently to an observation on the first day, but then correctly no change in snow depth is calculated for weeks, the error of the first day propagates for weeks, although the model only simulated the accumulation differently on the first day.

To visualise this behaviour, we present the temporal evolution of snow depth during the winter season 2005/2006 at five selected locations (Fig. 5). In this season, continuous snow cover with several freezing and thawing cycles caused roof
collapses and building evacuations in Bavaria (Strasser, 2008a). The time series show the temporal variability, the propagation of deviations, and the variability between models.

In the further evaluation, 31 of these time series for 83 locations are condensed into different evaluation measures. We try to evaluate the simulations based on different impact-relevant measures.


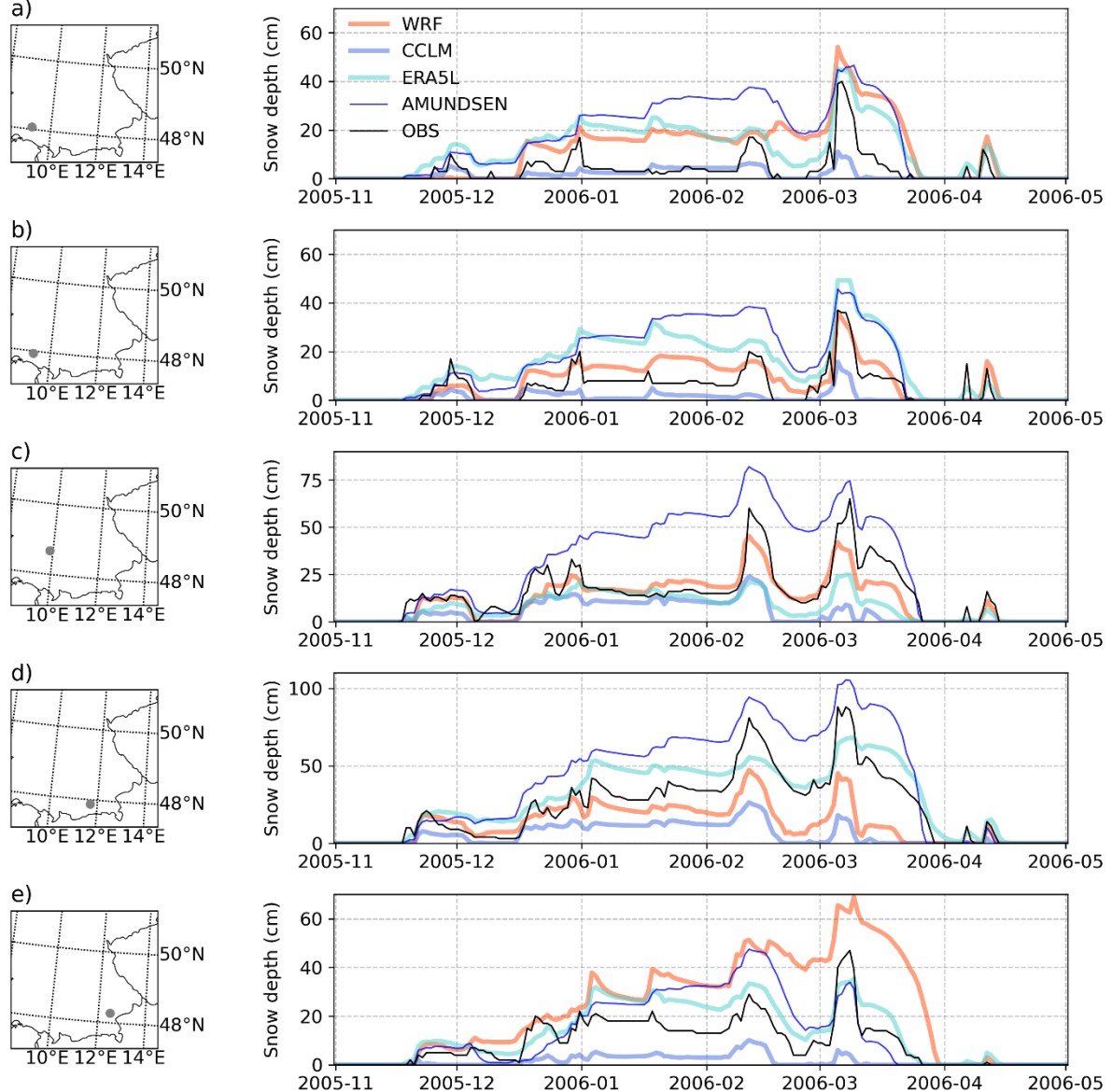

Figure 5: Observed and simulated daily snow depth over the winter season 2005/2006 at five locations.

The mean snow depth over the 31 winter half years in 1987 – 2018 varies largely between the models (Fig. 6). The deviations
to the climate stations are minor in the flat areas with generally less snow depth. WRF, ERA5L, and AMUNDSEN slightly

overestimate snow depth in the flat areas, while CCLM slightly underestimates. In the northern topographically complex regions of the middle mountain range Ore Mountains (far north-east), Thuringian Forest (north central) and Rhön (north at 10°E), WRF, CCLM, and ERA5L underestimate (red dots in the north in Fig. 6a-c), whereas AMUNDSEN overestimates snow depth. In the Alps, ERA5L largely overpredicts the snow depths, whereas CCLM underestimates heavily. WRF

underestimates to a moderate degree and AMUNDSEN shows almost no bias. A general overview of simulated and observed snow depths against the elevation is provided in Figure 7.

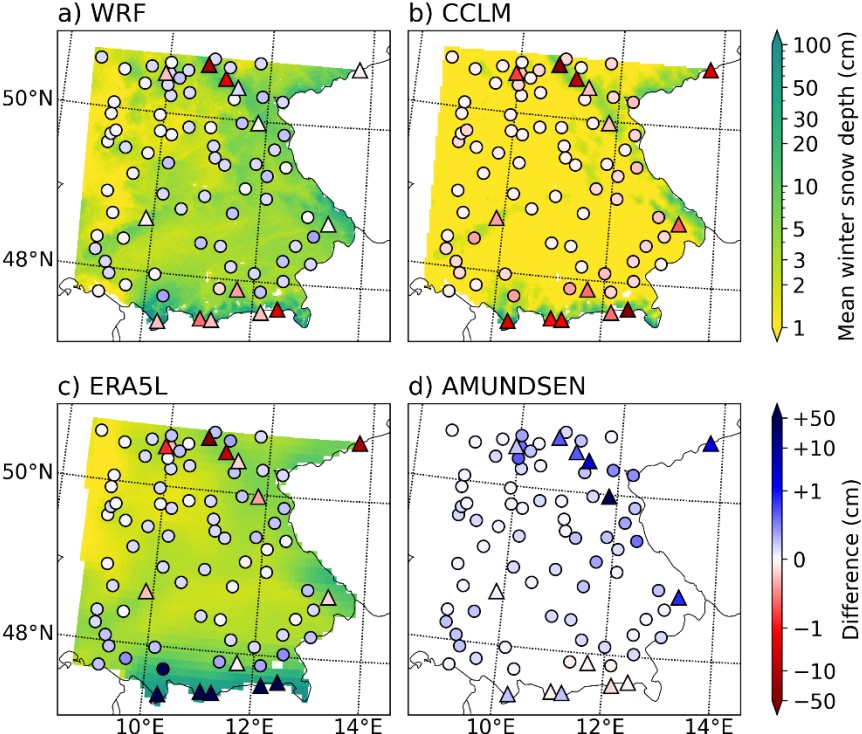

**Figure 6: Mean winter snow depth 1987 – 2018 simulated by the WRF model (a), CCLM (b), ERA5L (c), and AMUNDSEN (d). The**

**differences at the stations (coloured dots) are calculated as model minus observation. Red (blue) colour refers to an under-**
**(over)estimation of the model. Note the logarithmic colour scaling.**

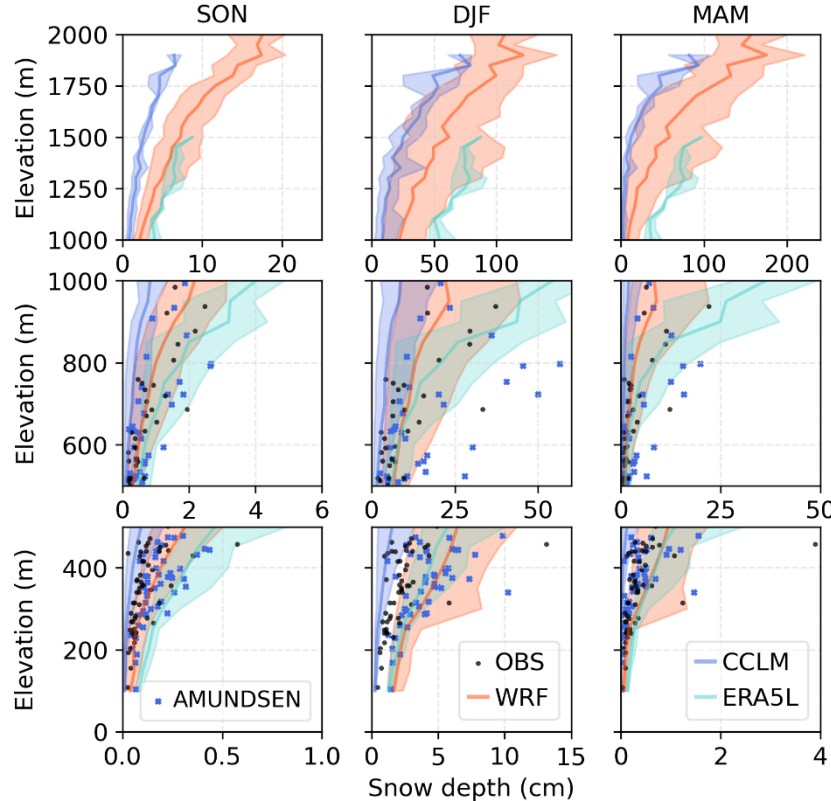

**Figure 7: Seasonal mean snow depth averaged over 1987 – 2018 compared over the whole range of elevations in the study area. Seasons refer to September – November (SON), December to February (DJF), and March to May (MAM). The transparent areas show the ranges of all grid cells at the respective elevation, and the line represents the median. Note the different x-axis scaling. Note that the elevation for AMUNDSEN stems from the CCLM, which is why it differs from the elevation of the observations.**

When assessing mean winter snow depth separately for each season and location, the WRF model shows low bias and errors (Fig. 8a, 8e). The CCLM can score good metrics for the stations less affected by snow (Fig. 8b), however systematically underestimating snow depth at all stations. This negative bias of the CCLM manifests itself in the fact that an overestimation of snow depth is simulated for almost no station at any season (Fig. 8b, 8f; the magnitude of the bias is almost equal to the MAE). The ERA5L simulations have the highest bias and errors, with the biggest deviations over complex terrain (Fig. 8g). Especially during snow-rich seasons, strong overestimation is present for the Alpine stations, whereby the stations in mid mountain ranges are heavily underestimated. AMUNDSEN shows the best overall rank correlation and the best reproduction of observed high snow depth seasons, however with a tendency to overestimate.

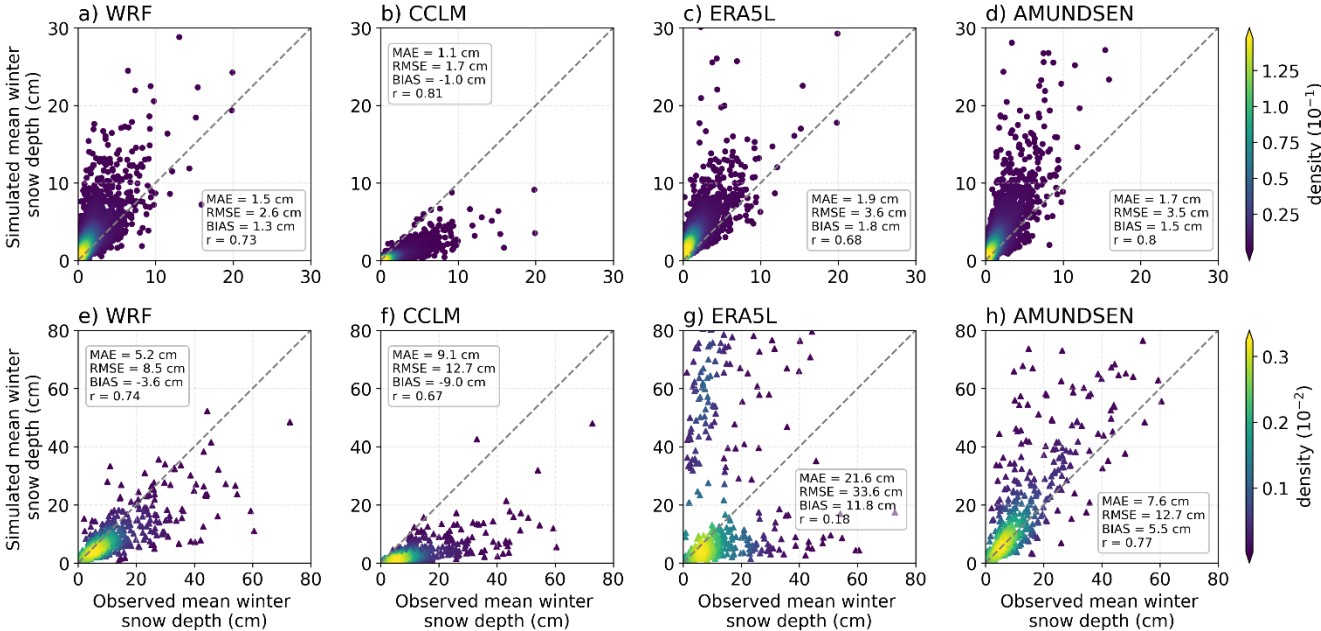

Figure 8: Mean winter snow depth for each winter season and location in 1987 – 2018. The upper row (a-d) refers to locations with less than 5 cm mean winter snow depth (dots in Fig. 6), whereas the lower row shows locations above this threshold (triangles in Fig. 6).

### 3.3 Snow duration and cover

Mapping the number of snow days averaged for 1987 – 2018 shows similar spatial patterns as the mean snow depth (compare Fig. 9 to Fig. 6), which also applies for the differences to the observations. CCLM generally underestimates the duration (Fig. 9b), whereas ERA5L generally overestimates apart from four locations in the northern middle mountain ranges (Fig. 9c). WRF and AMUNDSEN show lower differences with a similar spatial pattern (Fig. 9a and 9d).

The separate evaluation of each year is presented in Figure 10, where WRF and AMUNDSEN show a strong rank correlation and comparably low error metrics. Despite a high inter-seasonal and inter-station correlation, the CCLM simulation predicts on average 16 snow days less than observed at the low-lying localities (Fig. 10b) and 37 snow days less at snow-rich localities (Fig. 10f). ERA5L on the opposite simulates the duration 29 - 35 days too long (Figs. 8c, 8g).

The snow cover fraction during December to February averaged for 2000 – 2018 amounts to 42% based on MODIS (see Fig. 11). The CCLM strongly underestimates the snow cover in the whole study area (20%) resulting in less than half of the remote sensed fraction. ERA5L shows an overestimation in the southern Alpine part and slight underestimation in the remaining study area (Fig. 11c) amounting to 37% over the whole study area. For the WRF simulation, this parameter is not openly available.

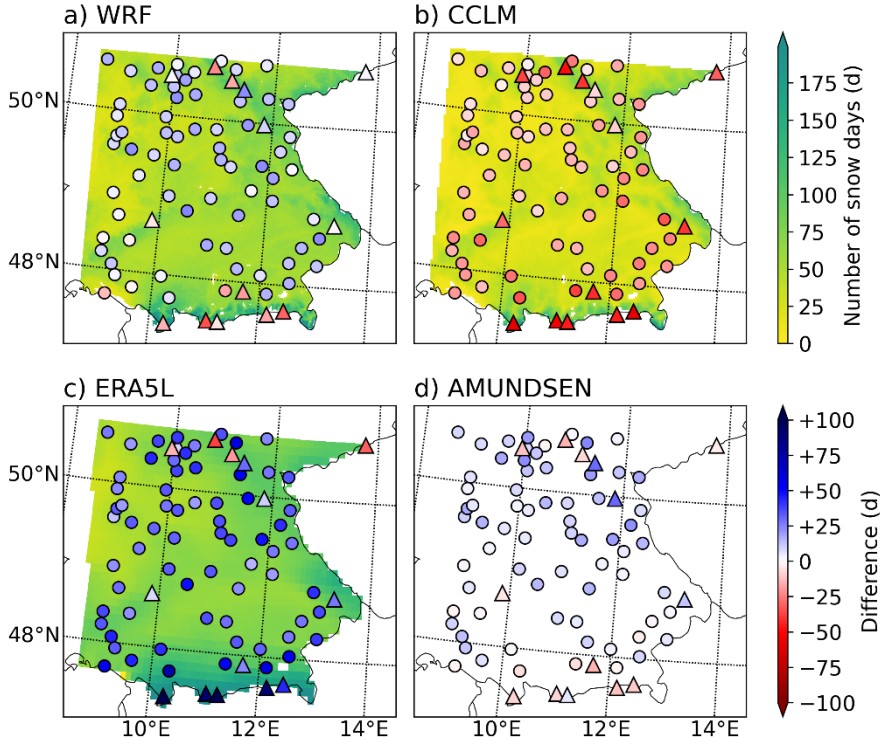

370

**Figure 9: Mean annual number of snow days during 1987 – 2018 simulated by the WRF model (a), CCLM (b), ERA5L (c), and AMUNDSEN (d). Snow days are defined as days with more than 1 cm snow depth. The differences at the stations (coloured dots) are calculated as model minus observation. Red (blue) colour refers to an under- (over)estimation of the model.**

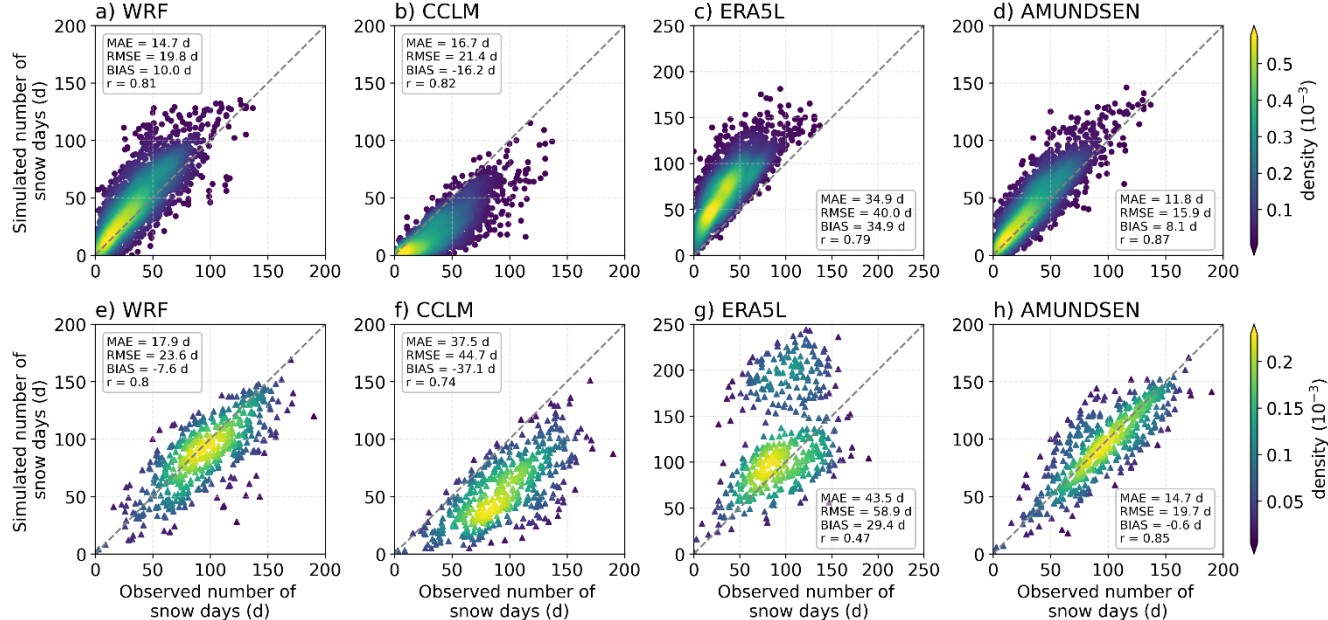

**Figure 10: Number of snow days for each year and location in 1987 – 2018. The upper row (a-d) refers to locations with less than 5 cm mean winter snow depth (dots in Fig. 9), whereas the lower row shows locations above this threshold (triangles in Fig. 9).**

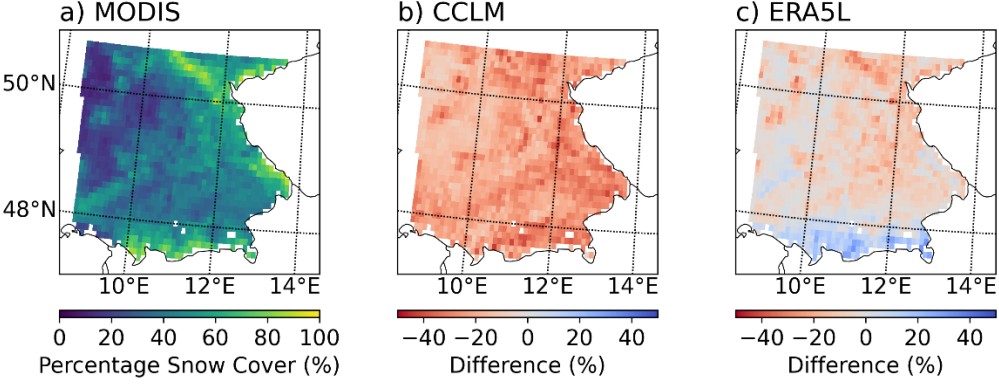

**Figure 11: Mean snow cover fraction during DJF averaged over the period 2000-2018. For the remote sensing product (a, MODIS MOD10C1.061), only data with at least "okay" quality are considered. The differences (b,c) are calculated as model minus remote sensing product, where red (blue) colour refers to an under- (over)estimation of the model. Extended winter season (November to April) is shown in Fig. S2.**

### 3.4 White Christmas and short periods

The tendencies for over- and underestimation of mean winter snow conditions (sections 3.2 & 3.3) also manifest when assessing shorter time periods than seasonal averages. The percentage of white Christmas during 1987 – 2018 results in four

very different predictions. The spatial pattern follows the previous evaluations of mean snow depth and snow cover duration (compare Fig. S3 to Figs. 6 and 8). Averages over all seasons and locations are provided in Table 3. There, the CCLM and AMUNDSEN outperform WRF and ERA5L with Matthew's Correlation Coefficients of 0.67 and 0.66, however with the tendency to underestimate (CCLM) and overestimate (AMUNDSEN) the occurrence of white Christmas.

**Table 3: Percentages of white Christmas averaged over all 1987 – 2018 and all locations in the study area.**

|  | WRF | CCLM | ERA5L | AMUNDSEN |
|---|---|---|---|---|
| Correct prediction | 76.3 % | 89.0 % | 69.3 % | 84.7 % |
| …thereof white Christmas | 19.8 % | 14.1 % | 22.4 % | 20.9 % |
| …thereof non-white Christmas | 56.5 % | 74.9 % | 46.9 % | 63.8 % |
|  |  |  |  |  |
| False positives | 20.6 % | 2.2 % | 30.2 % | 13.3 % |
| False negatives | 3.1 % | 8.8 % | 0.5 % | 2.0 % |
| MCC | 0.51 | 0.67 | 0.49 | 0.66 |

As white Christmas refers to only one selective time window of the year, we furthermore assess all 3-day moving windows over the extended winter season. Thereby, the MCC scores slightly improve (see MCC_3d in Tab. 4).

As a proxy for the opportunity to do cross-country skiing, we assess 5-day moving windows, during which the snow depth of all 5 days needs to be above 10 cm. This analysis is only carried out at the locations, where the mean snow depth is above 5 cm (see triangle markers in Fig. 1). For these conditions, the MCC scores of the CCLM and ERA5L drop, whereas the MCC of WRF and AMUNDSEN amount to 0.62 and 0.69, respectively (MCC_XC in Tab. 4).

**Table 4: MCC scores for 3-day moving windows with at least one day, where snow depth is above 1 cm averaged over all 1987 – 2018 and all locations in the study area. MCC scores for 5-day moving windows with snow depths above 10 cm for all 5 days.**

|  | WRF | CCLM | ERA5L | AMUNDSEN |
|---|---|---|---|---|
| MCC_3d (3 days, where any day > 1 cm) | 0.64 | 0.66 | 0.59 | 0.70 |
| MCC_XC (5 days, where all 5 days > 10 cm) | 0.62 | 0.46 | 0.36 | 0.69 |

## 3.5 Evaluation of extreme snow depths

Figure 12 provides the simulated and observed maxima of snow depth, where the WRF model shows small deviations and almost no bias at low-lying localities (Fig. 12a). Observations at snow-rich locations are slightly underestimated (Fig. 12e). The general underestimation of CCLM also applies for the maximum snow depths (Figs. 12b, 12f). The strong negative bias suggests that the model is not appropriate to assess extreme snow depths, as already described in the documentation by Doms

et al. (2021). The rank correlations of r = 0.79 and 0.71 however suggest, that the model is able to well reproduce inter-site and interannual differences. ERA5L can reproduce annual extreme snow depths at low-lying stations even slightly better than WRF (Fig. 12c), but tends to overestimate extreme snow depths yielding the highest RMSE and lowest rank correlation over complex terrain (Fig. 12g). The largest deviations occur for snow depths above 50 cm in complex topography (Fig. 12g). The difference of the mean annual maximum snow depth for the 83 stations correlates strongly with the elevation bias of ERA5L (see Figs. 4c, 4f, and Tab. S3; r = 0.91). We argue that the too low spatial resolution is a major contributor to the deviations of extreme snow dynamics in the middle mountain ranges and Alps. For the other models, no significant correlation is found for this dependence. AMUNDSEN slightly overpredicts extreme snow depths with a moderate positive bias for all stations.

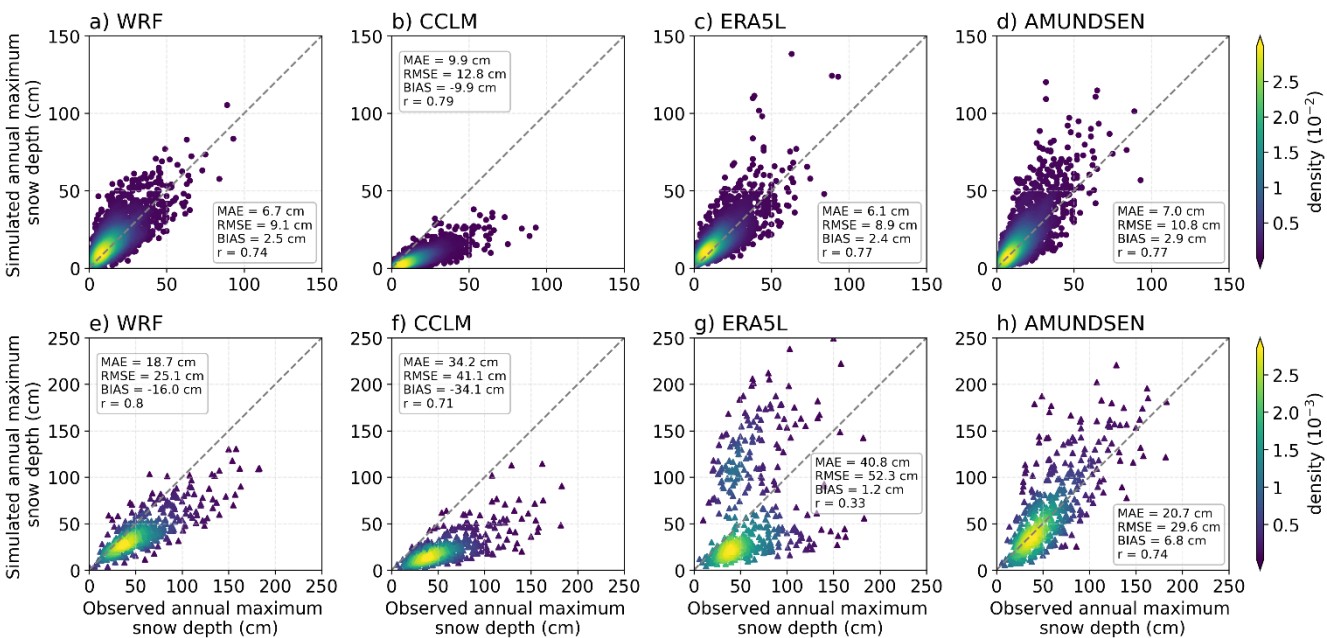

**Figure 12: Annual maxima of snow depth for each year and location in 1987 – 2018. The upper row (a-d) refers to locations with less than 5 cm mean winter snow depth, whereas the lower row shows locations above this threshold.**

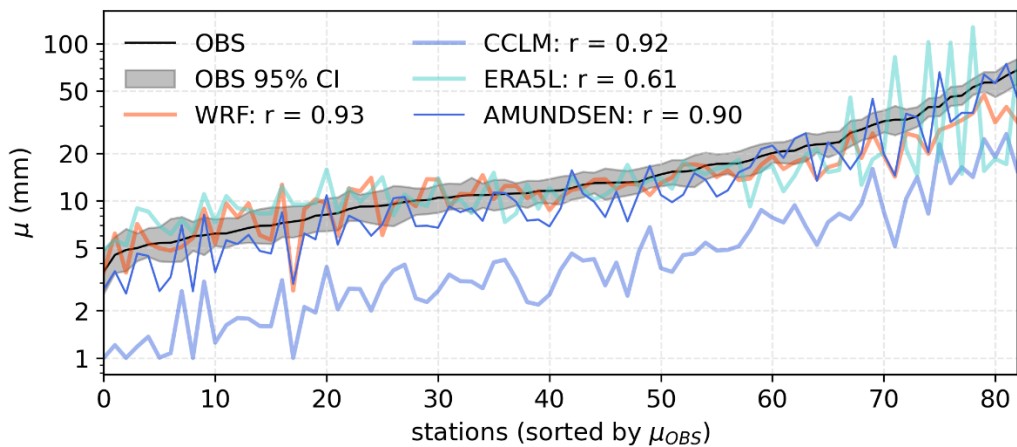

**Figure 13: Fitted GEV location parameters for the observed and simulated annual maxima of snow depth. Note the logarithmic scaling of the y-axis.**

We further assess, if the GEV location parameters µ fitted on the modelled annual maxima of snow depth are within the 95 % confidence interval of the respective observation-based µ for each station (see Figure 13). We find that the CCLM is not able to reproduce this extreme value statistical property within the observational range at any station. AMUNDSEN shows the highest agreement with 57% of stations, followed by WRF (51%) and ERA5L (34%). ERA5L cannot capture the rank correlation between the stations (r = 0.61) as well as the other models (r > 0.91) due to larger deviations at snow-rich localities.

The WRF model shows good agreement for stations with location parameters below 20 mm but underestimates the upper range (Fig. 13).

The seasonality of the annual maximum snow depth is visualised as bivariate kernel density estimation (Figure 14). The general timing of annual maximum snow depths is reproduced well by all models (compare the marginal histograms in Fig. 14), whereby the CCLM simulates not enough snow depth maxima after February (Fig. 14b). For the comparison of each station-

year combination in 1987 – 2018, the WRF model (Fig. 14a) shows the highest agreement with low bias.

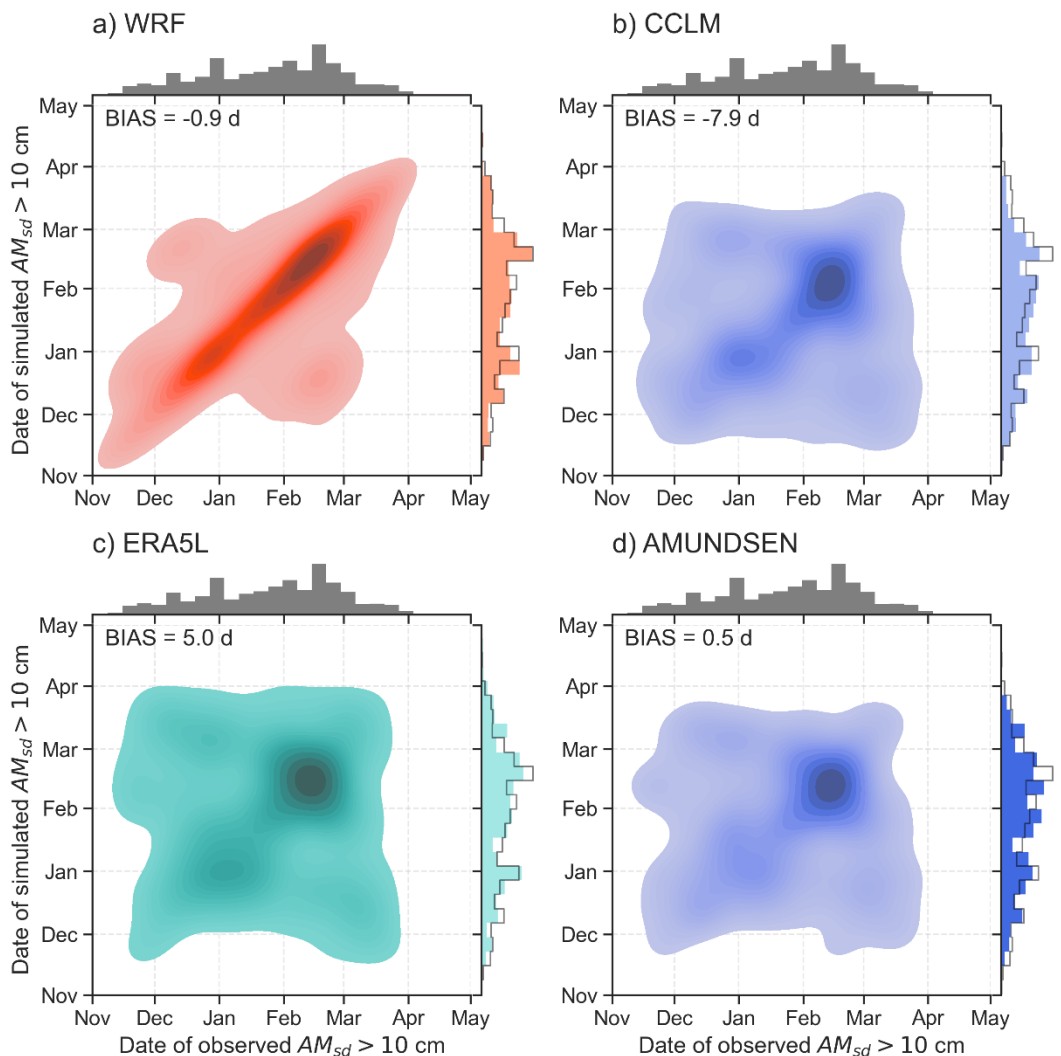

**Figure 14: Bivariate kernel density plot of dates for annual maxima snow depth above 10 cm for each year and location in 1987 – 2018 simulated by the WRF model (a), CCLM (b), ERA5L (c), and AMUNDSEN (d). The marginal distributions aggregated over all years and stations are given as histogram aside, where the grey histogram and grey line histogram refer to the observations.**

## 4 Discussion of uncertainties

The representation of snow dynamics is important for impact assessments (see Section 1) and for the coupling within the regional climate model in case of the WRF and CCLM.

Depending on the measure of interest, the evaluation yields very different results, even though all simulations are driven by the same large-scale atmospheric conditions from ERA5. This indicates the large model uncertainties regarding snow dynamics. These uncertainties result from different sources and are interconnected, which is why they are difficult to disentangle (Monteiro and Morin, 2023).

In addition, the comparison between point in-situ measurements and gridded simulations is affected by the spatial variability of snow depth (Clark et al., 2011). The standardised local environment of the climate station might not capture the landcover and topography of the surrounding area, which in turn governs the gridded simulation (Meromy et al., 2012). The according deviations are expected to be higher for coarser model resolution, complex topography, in areas covered by forest vegetation, and in open areas prone to wind redistribution (Mortimer et al., 2020).

Furthermore, the rather low station density leads to unsatisfactory coverage in parts of the study area. However, the stations were selected according to their low amount of missing data, and they represent well the range of elevations between 150 m and 1000 m (Fig. 2). Additionally, the analysis was supplemented by satellite data from MODIS to ensure spatial representativeness.

    In Table 2, the significant correlation between elevation bias and the biases of temperature and precipitation for ERA5L

indicates that the spatial resolution of the model contributes considerably to the deviations. Due to the 9 km grid cell size, the complexity of the terrain cannot be reproduced in parts of the study area.

    For the WRF simulation, a significant correlation between elevation and temperature bias is found indicating that the lower elevated stations show a more pronounced cold bias. Overall, the WRF shows a systematic underestimation of temperature. Collier and Mölg (2020) refer this behaviour to a miscalculation of the mean grid cell albedo in the NOAH_MP (Tomasi et

al., 2017). Generally, it should be emphasised that the albedo of exposed snow surfaces differs from snow over surfaces with shrub or forest vegetation. The areal simulations of WRF, CCLM, and ERA5L represent mean grid cell values and depend on the land cover parametrisation of the respective grid cell. At the point of the climate station, exposed snow areas predominate due to the standardised environmental conditions with grass cover. Hence, the comparability is limited for grid cells, where the model assumes forest. Furthermore, the temporal variability of snow and albedo at the point scale can be higher than for

areal averages (Clark et al., 2011). Remote sensing products provide observation-based areal estimations for the albedo. We compare the albedo of the WRF, CCLM, and ERA5L simulations to the remote sensing product MODIS MCD43C3 at 0.05° resolution (Schaaf and Wang, 2021) for the whole study area without the Alps (Fig. 15a) and the Alps only (Fig. 15b) and confirm the strong overestimation within the WRF simulation.

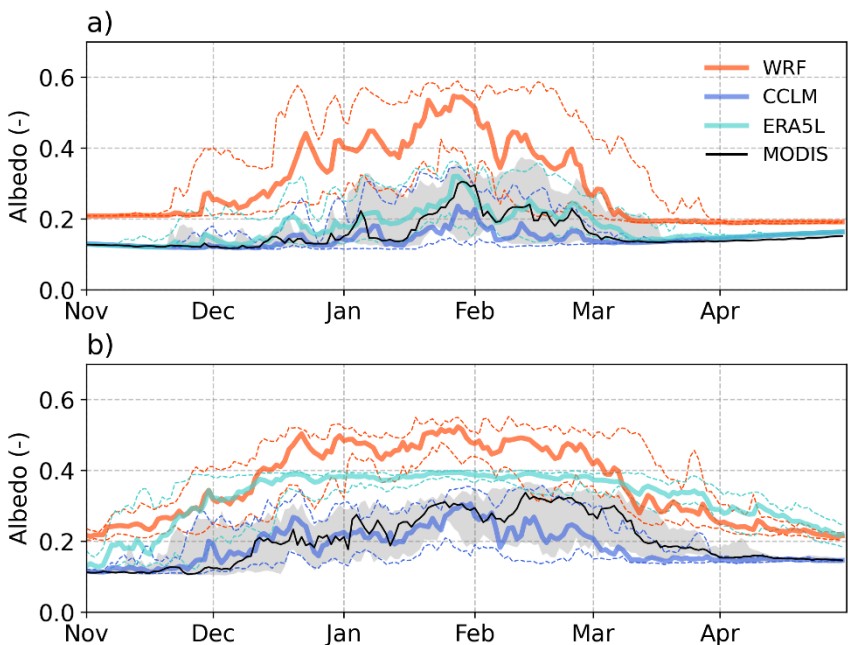

**Figure 15: Albedo for the years 2000 to 2018. Daily albedo values are averaged over the whole study area (a) without the Alps and (b) for the Alps (south of 47.8 °N and west of 10° E) only. The thick lines denote the median over all seasons. The area between the dashed lines and the grey transparent area show the inner 50% range. For the remote sensing product (MODIS MCD43C3), only data with at least "mixed" or good quality are considered, where at least half of the area needs to be covered with valid values.**

ERA5L agrees over the non-alpine area, however overestimating in the Alps, where the strongest deviations occur during early and late winter reflecting the huge positive bias in the snow cover duration in the Alps (Fig. 9c). As ERA5L is an offline simulation driven by the climate of ERA5, the simulated albedo of ERA5L has only very minor effects on the temperatures shown in Figure 3. The empirical parametrisation in CCLM agrees with MODIS during the first months (until mid-January and beginning of February, respectively), however underestimates the albedo afterwards. This is due to the underestimation of

snow cover duration (Figs. 9b, 10b, 10f). As the land surface scheme TERRA_ML is run coupled with the atmospheric simulation, the albedo affects the simulated air temperature. However, the low biases in temperature (Fig. 3) suggest that the overall representation of the albedo in TERRA_ML does not translate to strong discrepancies in temperature of CCLM.

The different albedo simulations are compared over one selected winter season and five locations (Fig. S5, Tab. S6). There, the negative temperature bias of the WRF simulations (Fig. 3) occurs mainly during periods with snow and therefore

overestimated snow albedo (Fig. 16) confirming the assumption by Collier and Mölg (2020). Liu et al. (2021) propose a modified snow albedo scheme for the NOAH_MP with on average lower albedo values than the default scheme improving the representation of sensible heat flux, air temperature, and snow depth over the Tibetan Plateau. The albedo scheme of AMUNDSEN for exposed snow surfaces yields a similar albedo estimation as the WRF simulation for non-forested grid cells

(Fig. S5). However, AMUNDSEN represents the point scale and is not run coupled with the atmospheric modelling therefore
not affecting the modelled climate.

Furthermore, the snow surface albedo also governs the energy budget of the snow layer (Essery et al., 2013). The slight overestimation in mean snow depth and snow cover duration of the WRF simulations can be attributed to the biases in albedo and temperature. Applying different albedo parametrisations induces considerable differences in the simulated snow depth and snow cover duration. For the WRF simulations, 17 of the 83 locations are classified as forest. The mean bias of winter snow
depth and snow cover duration over all stations amount to +0.4 cm and +6.8 d (Figs. 8 and 10). For the 17 forest grid cells, however, the deviations amount to -0.9 cm and +3.2 d, whereas the remaining non-forested locations show larger overestimations of snow dynamics (+0.9 cm and +7.7 d).

Even though ERA5L simulates a lower surface albedo over non-forested areas than WRF, it shows the strongest positive bias in mean snow depth and snow cover duration. The largest overestimations are located in the (pre-)alpine areas in the south of
the study area, however, the snow cover duration is overestimated for almost the whole region. Daloz et al. (2022) also report positive deviations of ERA5L in the extent, fraction, and duration of snow cover over Europe compared to MODIS. Kouki et al. (2023) find in an evaluation with various satellite-based data that ERA5L also overestimates the SWE in the Northern Hemisphere in spring. Monteiro and Morin (2023) confirm these findings over the Alps, where ERA5L shows the strongest positive bias in snow depth and snow cover duration of all intercompared models at all elevation ranges. When developing the
new snow scheme for the next generation ECLand model (Bousetta et al., 2021), these deviations should be accounted for.

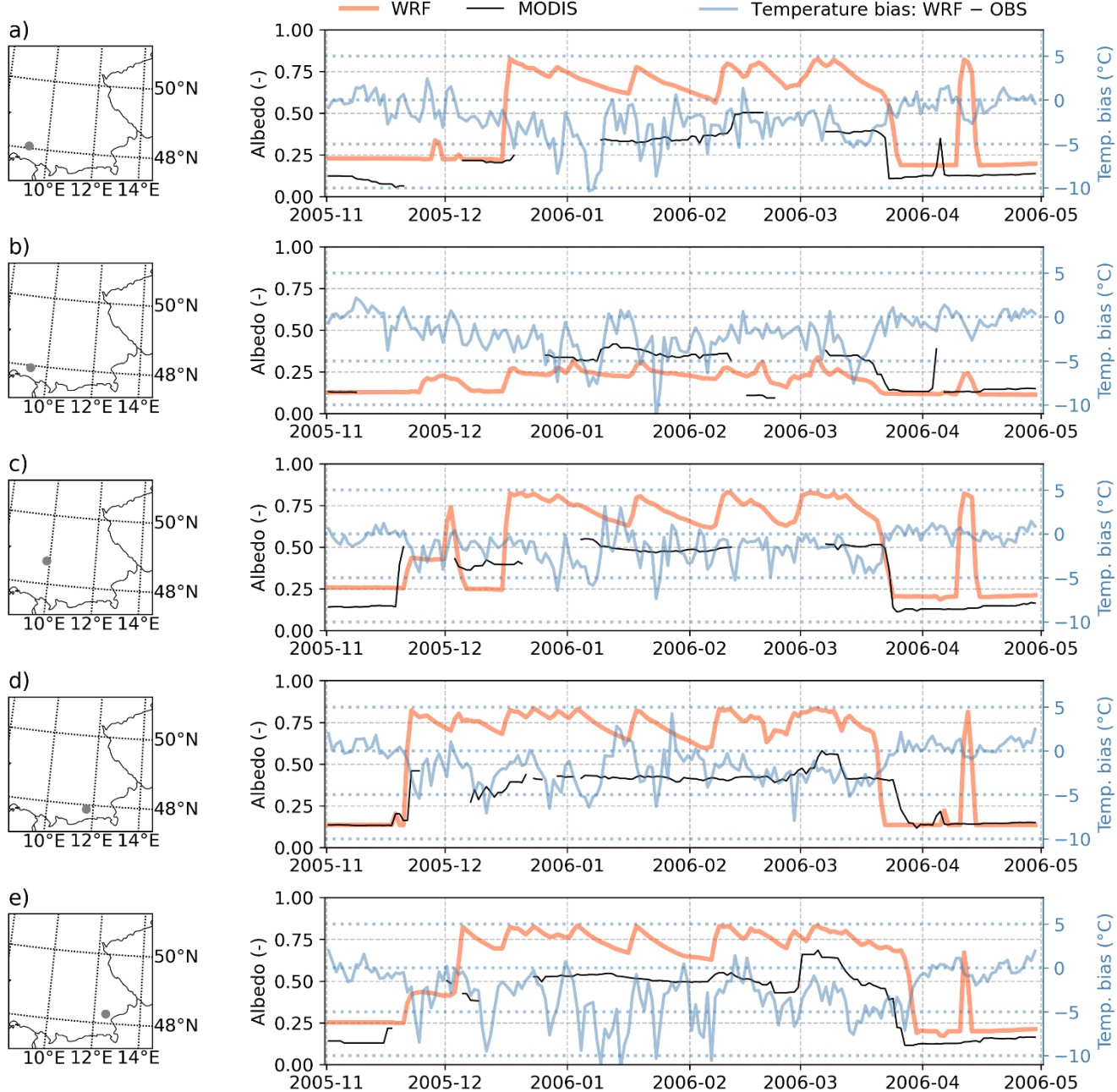

**Figure 16: Daily albedo based on remote sensing (MODIS MCD43C3) and the WRF simulation over the winter season 2005/2006 at five locations. For the MODIS time series, only data with at least "mixed" or good quality are shown. The second y-axis shows the temperature bias of WRF compared to the weather stations. The vegetation type of the respective WRF grid cell is grassland (a, c, e), evergreen needleleaf forest (b), and urban and built-up land (d) affecting the simulated albedo.**

Despite the skilful reproduction of precipitation and air temperature (Fig. 3b and 3e) and realistic ranges of simulated albedo (Figs. 15 & S5), the CCLM simulates snow depth, the number of snow days, snow cover fraction, and snow depth extremes with large negative biases. In contrast to our analysis, Lüthi et al. (2019) diagnosed a slight overestimation of SWE over Switzerland for a similar model setup, i.e. employing COSMO-CLM at 2.2 km with the TERRA_ML land-surface model driven by ERA-Interim (Ban et al., 2014). The CCLM and TERRA_ML setup analysed in this study however suggests that it is not appropriate for any further impact analysis.

Based on the same climatic conditions of CCLM, AMUNDSEN tends to slightly overpredict snow depth and snow cover duration. The larger overestimations of the AMUNDSEN simulation are located in the middle mountain ranges (Fig. 6d and 9d). Especially during snowy winter seasons, AMUNDSEN produces snow depths that are heavily overestimated. An exploration of these locations reveals that the CCLM has a general positive precipitation bias there, which is partly diagnosed by the rank correlation of r = 0.32 between elevation and precipitation bias. However, this positive bias mostly applies for the stations above 500 m elevation north of 48.5° N (Fig. S7).

Apart from the sources of uncertainties discussed above, also the parametrisation of snow density adds uncertainty (Essery et al., 2013). Density of newly fallen snow is usually in a range of 20 kg/m³ to 300 kg/m³ and depends on the air temperature inducing dry to wet snow characteristics and the wind speed (Lee et al., 2023). Here, all models parametrise the density of fresh snow differently depending on temperature, where only ERA5L additionally includes near-surface wind speed. Also, compaction and metamorphism are parametrised differently, where three models (WRF, ERA5L, AMUNDSEN) follow Anderson et al. (1976) for the description of compaction. This semi-empirical scheme is widely used in snow and land-surface models and assumes a two-stage compaction due to metamorphism and pressure from above snow mass (Aschauer et al., 2023). It further employs viscosity coefficient dependent on temperature to model stress-induced compaction. With wet snow in the respective snow layer, settling of the snowpack is described. Even though this approach is quite sophisticated, not all processes leading to densification are captured, which may partly contribute to deviations of modelled and observed snow depth. Koch et al. (2019) note that rain-on-snow events or periods of warm weather may cause such deviations, which are expected to occur in the elevation range of our study area. Hence, the estimation of snow density contributes to inter-model differences and also induces uncertainty for the evaluation of snow depth. At the climate stations however, snow density or SWE is not measured and can therefore not be evaluated.

In addition, the number of considered snow layers is found relevant for the reproduction of snow depth and cover (Arduini et al., 2019; Jin et al., 1999). Jin et al. (1999) emphasise that multiple snow layers improve the representation of temporal variability within diurnal and seasonal time scales. Xue et al. (2003) highlight the importance of multiple snow layers for the ablation period, as multiple layers are able to separate soil temperature and surface temperature. This is considered beneficial for reproducing the variability of snow surface temperatures, while single-layer setups tend to simulate surface temperatures around the freezing point with little variability. Here, the multi-layer setups of WRF and AMUNDSEN show a better performance of the snow cover duration, timing and intensity of annual maximum daily snow. The new five-layer snow scheme of ECLand (Arduini et al., 2019) improves the simulation of melting periods compared to the single-layer scheme of ERA5L.

The evaluation of simulations for short periods within the year is very relevant for tourism related topics. For the prediction of white Christmas, snow during any 3-day period, and snow consistently above 10 cm during a 5-day period, the AMUNDSEN can achieve the highest classification scores. All models perform with MCC in a range from 0.36 to 0.70, indicating a moderate to strong positive relationship between the model classification and the observation. Still, the tendency to classify a period as "snowy" follows the general behaviour of the respective model to overestimate or underestimate snow depth.

For the representation of extremes in snow depth the differences between the point scale and areal simulations may affect the comparison. Averaged over all locations, one would expect a slight underestimation of simulated annual maximum snow depth as the climate station shows an exposed snow surface, whereas grid cells may represent mixed surfaces or forested areas. Furthermore, the point scale shows a higher temporal variability than areal averages therefore leading to more pronounced extremes. Still considering these limitations, the assessment reveals that WRF, AMUNDSEN, and ERA5L can well reproduce snow depth extremes in low-lying areas, whereas WRF and AMUNDSEN can clearly add value to the representation of extremes over complex terrain compared to ERA5L.

## 5 Conclusions and recommendations

In conclusion, the high-resolution climate model WRF and the hydro-climatological model AMUNDSEN driven by CCLM can add value compared to the state-of-the-art reanalysis product ERA5L. The CCLM shows a skilful reproduction of the climate but systematically underestimates all snow dynamics. Based on this assessment, one can draw some recommendations for model application and further model development.

First, simulations of snow dynamics have to be carefully evaluated according to the further intended use of the data. As other studies, such as SnowMIP (Essery et al., 2013), this evaluation has confirmed that the setup of the regional climate model or land surface model can largely influence the snow depth simulations, even if they are all driven by the same large-scale atmospheric conditions. Here, we have shown that this variability also occurs at lower elevation ranges (150 m to 1000 m). With coarse-resolution climate models, the complexity of the terrain often prevents comparability between the results of the climate model and the observations. For the 83 locations in southern Germany with a terrain with different degrees of complexity, the resolution of the high-resolution RCMs is sufficient to reproduce the elevations with moderate to low deviations. For the representation of the winter climate, high-resolution RCMs can add value compared to ERA5L. This translates also to a benefit for the representation of snow dynamics, except if the snow model is not appropriately parametrised for the study area (as in the case of CCLM and TERRA_ML). Hence, for a study area with medium-complex topography, we can generally recommend to further use the snow depth from a high-resolution RCM snow scheme. In case of a well-represented climate but strong biases in the snow simulation (as here for the CCLM), the added value of the high-resolution climate simulation can be utilised by setting up a separate snow model (as AMUNDSEN in this study), where a calibration to the local conditions might further improve the reproduction of observed snow depth. Hence, for local and regional snow impact assessments, high-resolution climate models can be a valuable tool. However, we recommend to analyse climate and snow

biases carefully. In order to fully make use of this potential in high-resolution simulations, it would be advisable to store and
provide not only simulated snow depth, but also SWE and fraction of snow cover.

Second, for future climate model and land surface model development, the evaluation in this study has revealed some potential for improvement. A thorough review or revision of the TERRA_ML snow scheme in the ICON-CLM model is recommended. For the NOAH_MP, the problem of too high albedos has been addressed by Tomasi et al. (2017) and the resulting cold bias during winter as well as overestimation of snow dynamics are confirmed for southern Germany in this study. New albedo
parametrisations are already proposed (Liu et al., 2021) and are to be evaluated. The overprediction of mean snow depth and snow cover duration in the ERA5L simulations partly result from the elevation bias and according temperature bias. However, the general tendency to overestimate snow cover duration and the underestimation of extreme daily snow melting indicate that the snow scheme needs to be revised accordingly in the course of the development of the new ECLand model.

Generally, parametrisations and snow models tend to be developed and evaluated disproportionately in alpine and arctic areas
(e.g. Essery et al., 2013, Krinner et al., 2018, Liu et al., 2021). This is understandable as snow dynamics have high relevance there and long records without non-natural influences are available. However, considering the relevance for snow-related impacts, it is also important that snow dynamics in more densely populated areas are well represented in the simulations. Hence, we propose the study area in southern Germany as testbed for further investigations such as the LUCAS phase 3 (Daloz et al., 2022).

On the other hand, high-resolution climate model or earth system model simulations could further improve our knowledge of snow dynamics in topographically highly complex regions, where state-of-the-art reanalysis data and earth system models show large deviations from observations (Daloz et al., 2020; Kouki et al., 2022). However, observations are sparsely distributed in these regions. Snowfall and snow storage play an important role in freshwater availability, with strong implications of rising temperatures (Simpkins, 2018). Hence, high-resolution models could support decision-making regarding water infrastructure
design and management.

Lastly, we recommend to evaluate the performance of snow models regarding extreme snow dynamics. Possible measures are proposed in this study, such as daily maxima of snow depth and their seasonality. They could be extended by assessing daily maxima of snow melt and accumulation analysing the SWE. Our knowledge about future snow dynamics depends on model simulations. Due to the high impacts of extreme snow conditions on the human society, it is relevant to capture these events
in the simulations.

*Data availability.* The ERA5-Land data are available from https://doi.org/10.24381/cds.e2161bac (Muñoz-Sabater et al., 2021), the WRF simulations are stored at https://doi.org/10.17605/OSF.IO/AQ58B (Collier, 2020). The COSMO-CLM simulations are available from https://dx.doi.org/10.5676/DWD/HOKLISIM_V2022.01 (Brienen et al., 2022). The
observational data are provided by the German Weather Service at https://opendata.dwd.de/climate_environment/CDC/observations_germany/climate/daily/kl/historical/. MODIS snow cover

fraction is available from https://doi.org/10.5067/MODIS/MOD10C1.061 (Hall and Riggs, 2021) and MODIS albedo data are available from https://doi.org/10.5067/MODIS/MCD43C3.061 (Schaaf and Wang, 2021).

*Author contributions.* BP conceptualised the study, performed the simulations, and conducted the formal analysis. ASD contributed to the conceptualisation. BP prepared the original draft with significant contributions from ASD.

*Competing interests.* The author declare that they have no conflict of interest.

*Acknowledgements.* We cordially thank all data provisioners, which are cited in the main article: Brienen et al. (2022), Collier (2020), German Weather Service, and Muñoz-Sabater et al. (2021). The open source version of AMUNDSEN written in Python (Warscher et al., 2021; https://github.com/openamundsen/openamundsen) is cordially acknowledged. Further, we thank Susanne Brienen from the German Weather Service for information on the COSMO simulations and provision of the albedo data. This research has been funded by the Deutsche Forschungsgemeinschaft (DFG, German Research Foundation) under
Germany's Excellence Strategy—EXC 2037 'CLICCS—Climate, Climatic Change, and Society'—Project Number: 390683824, contribution to the Center for Earth System Research and Sustainability (CEN) of Universität Hamburg.

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
