# Peer review of "Snow depth in high-resolution regional climate model simulations over southern Germany - suitable for extremes and impact-related research?"

_The Cryosphere, 2023_

## Author Comment (AC1)

Dear Reviewer,

We thank you for your constructive critique and the valuable comments and suggestions to improve the quality of the manuscript. We address your comments (dark grey) with our responses (blue) in the following. We think that the outlined additional analysis based on your recommendations will help to improve the study.

Poschlod and Daoz analyze snow depth from two high-resolution climate models and one reanalysis for an area in Southern Germany. The purpose of the study is not clear, since no research aims are stated. The title hints to "suitable for extremes and impact-related research?", however, most of the manuscript is just model evaluation and little on extremes. Some of the impact-related variables are highly questionable. There are some major concerns on parts of the methodology, and the manuscript needs a clearer structure before of publication quality.

**Major points**

- I miss a description of why the research setup. What are your aims and/or hypotheses? Why snow depth? It is not a state variable, and you never discuss how different density estimates might impact your results. Why in-situ observation and not remote sensing? Again, the mismatch of point-vs-grid could be highlighted more clearly, and also the impact of resolution, since I guess a point is more representative for the 1.5km cell than for the 9km cell.

Hypothesis / Aims:

Thank you for this comment. We will better clarify the hypothesis and aims. The main question of the study is: Can new-generation high-resolution regional climate models represent snow depth dynamics at high temporal (daily) and high spatial detail? One according hypothesis is: The high-resolution *dynamical* downscaling of ERA5 atmosphere (via CCLM & WRF) can add value compared to the state-of-the-art ERA5L reanalysis product. This aim is indeed related to "just model evaluation", however at high spatio-temporal scales. This is very relevant for impact research.

Further, we try to show the complexity of this topic by highlighting elevation bias, climatic driver biases, albedo, and land cover. We also touched upon the point-vs-grid mismatch (L405-409), but we will further enhance the discussion in that regard. We will also discuss snow density as uncertainty source (thank you for this hint).

Motivation / Relevance:

Impact research needs information about impactful events at the local scale. Climate change affects the dynamics and conditions, which is why observation-based analyses are limited. Often, coarse-resolution RCMs or even GCMs have been used to drive snow models at local/regional scale. However, bias adjustment, statistical downscaling and the de-coupling of the interactions of snow dynamics and climate (snow simulations do not feed back into the climate simulation) induce additional uncertainties and limitations.

The "new generation" of high-resolution RCMs could potentially directly provide snow depth information from their internal land surface / snow modules, which leads us to the guiding question: How good are they at representing snow depth?

Setup:

To answer the question and test the hypothesis, we need to:

1) explore high-resolution RCM simulations,
2) which cover not only single years but climatological periods (~ 30 years) to represent the variability and extremes
3) and which are driven by reanalysis in order to be able to compare to observations
4) define a baseline (in our case ERA5L)
5) define a reference (in our case in-situ observations)

1) – 3) strongly limits the choice of available simulations. The CCLM and WRF simulations are the simulations, which we found publicly available.

4) ERA5L as global land reanalysis is the state-of-the-art reanalysis product at 9km resolution, which is also driven by the same climate (ERA5 atmosphere) and therefore comparable.

5) For snow depth, in-situ observations are the typical validation reference (see e.g. https://tc.copernicus.org/articles/15/1343/2021/ ). However, based on your and the other reviewer's suggestion, we will add remote sensing data for snow cover validation of the gridded simulations.

In addition to 1) – 5), we added the AMUNDSEN simulations driven by CCLM at the point scale. This setup was added in the course of the evaluation, where the CCLM showed strong systematic underestimation of any snow variable, while representing the climate better than the other models (lowest biases and errors). Hence, driven by the perspective of impact research, we wanted to explore how the separate snow model AMUNDSEN can make use of the well representative CCLM climate.

- L210: In case of heavy snowfall, compaction can be of larger magnitude than melt. (Also how do you derive this index in case of missing observation data - you allowed 30%, right?) Then for maximum accumulation, snowfall might be a better variable than increment in snow depth. And for maximum melt, SWE change (which is anyway the state variable in the model). I suggest removing these analyses. If you want to focus on extreme accumulation and melt, which of course are variables with significant societal impact, then you need to choose appropriate variables. And if you want to use snow depth as proxy instead, at least you have to prove it is meaningful compared to snowfall and SWE loss. Currently, they cannot be trusted, and therefore should be removed from the manuscript.

We follow your suggestion and remove these analyses from the manuscript.

- Your study focuses on low-elevation snow cover, and your station coverage is rather limited. Especially, all stations are below 1000m. This is important to acknowledge in discussion and research setup.

We will emphasize the elevation of the stations in the Introduction and Discussion. Also, the station coverage will be discussed as uncertainty source.

- Consequently, your evaluations, as they are know, are heavily influenced by low snow amounts. For example, you can have high values of relative errors for irrelevant snow amounts. (this is less an issue for high-alpine sites, where you have large snow amounts, related L510). This needs to be discussed. Even better would be analyses that distinguish by elevation or snow amount, or summary metrics/table that take this into account.

Thank you for this suggestion. We will divide the evaluation of mean snow depth and annual maxima of snow depth (Figs. 7, 9, 11) into different categories (either bins of elevations or bins of mean annual snow depths). We will further provide metrics supporting this distinction.

- The discussion on snow albedo as driver of biases is good. However, the important figure is in the supplement and one that is of minor use in the main part.

We will add Fig. S9 to the main manuscript.

**Minor points:**

- L20: improves relative to what? Numbers are somewhat in between.

  Relative to CCLM, which is the driving climate for AMUNDSEN. We will clarify that.

- L23: "All models fail…" but observations do?

  In the case of white Christmas (snow depth > 1 cm), we would assume that observations can diagnose these conditions. The term "predict" might be misleading in this sentence. We will change to: "The presence or absence of white Christmas is reproduced with Matthew's Correlation Coefficients of 0.49 (ERA5L), 0.51 (WRF), 0.66 (AMUNDSEN), and 0.67 (CCLM)."

  Note: The use of the Matthew's Correlation Coefficient is discussed in later comments due to your suggestions.

- L25: what limitations?

  The still remaining deviations in intensity and seasonality. We will change "limitations" to "deviations".

- L26: Winter climate is more than just snow.

This statement was supposed to relate to precipitation and temperature and not snow. We will replace "winter climate" with "winter precipitation and temperature".

- L26ff: Not sure. Abstract numbers suggest high accuracy, in fact. (after reading the manuscript the low biases make sense, because you only look at low elevations) For climate change research, another important factor is the boundary forcing from GCMs.

  We will emphasize the low elevation already in the abstract and provide metrics tailored to the elevation ranges (see Major Comments).

- L66: Please be more specific. Blowing snow has already been implemented offline (10.5194/tc-15-743-2021) and also online (10.5194/gmd-16-719-2023).

  Thank you for these references. We are happy to add them.

- L79: Less means there is something, what?

  That was worded a too imprecisely: We haven't found any literature on snow depth extremes in climate models.

- Intro: Research aims are missing.

  see Major Comments; will be added.

- Intro: Also, I would have expected something on snow studies in climate models, such as 10.3390/atmos10080463, 10.1007/s00382-012-1545-3, 10.5194/tc-17-3617-2023, etc

  Thank you for these recommendations. We will add them to the introduction. Especially the extensive third study is very relevant.

- Table 1: would be easier to read if references were put in caption or similar. And ERA5-Land is not statistical downscaling.

  We will make the table easier to read (either putting references in the caption or a table footnote – whatever the journals layout allows). The font size of the template is larger than the final journal layout – so tables look a bit unproportioned here.

  The climatic drivers for ERA5-Land result from a linear interpolation (which can be called statistical downscaling). See https://doi.org/10.5194/essd-13-4349-2021: "ERA5-Land is driven by atmospheric forcing derived from ERA5 near-surface meteorology state and flux fields. The meteorological state fields are obtained from the lowest ERA5 model level (level 137), which is 10m above the surface, and include air temperature, specific humidity, wind speed, and surface pressure. The surface fluxes include downward shortwave and longwave radiation and liquid and solid total precipitation. These fields are interpolated from the ERA5 resolution of about 31 km to ERA5-Land resolution of about 9 km via a linear interpolation method based on a triangular mesh."

  Hence, the downscaling of the climate drivers from 31km ERA5 to 9km ERA5L is done via statistical interpolation.

- Sec 2.4: The number of snow stations is quite low, compared e.g. to (10.5194/tc-15-1343-2021). Did you take only stations which had snow, temp, and precip simultaneously? Also the maximum elevation is rather low… Since your study is about snow, maybe it would make more sense to include more observed snow data (not necessarily with temp and precip).

  We first downloaded all stations in the study area via the R package rdwd (as also Matiu et al. 2021 in 10.5194/tc-15-1343-2021), however with the constraint that their measurement extends longer than 1988 (starting winter season of our analysis). That yielded 408 stations. this number seems to be comparable to Matiu et al. However, we remove all stations with more than 30% of missing snow depth data during the period 1987 – 2018. This results in the selection of the subset of stations in the article. Using stations with shorter coverage would distort some of the applied metrics in the article.

- Fig4: I don't see any large dependence between temp and precip (except for ERA5L), so maybe if your point is temp/precip bias depend on elevation biases, it would be better to put elevation bias as continuous x-scale (and not discrete point shape, which makes it hard to read).

  Thank you for this suggestion. We will rearrange the figure as proposed.

- FigS2 is quite good and relevant, I suggest moving it into the main manuscript. Just make the elevation bins wider to reduce noise and have a constant line for ERA5L. Also elevation seems not to match between DWD and AMUNDSEN.

  We will include Fig. S2 in the main article. The elevations of DWD and AMUNDSEN do not match as the AMUNDSEN elevation stems from CCLM, as the CCLM simulation is used as climatic driver.

- Plots with obs vs. sim would benefit from a 1:1 line

  Will be added.

- L315: since your prevalence is 3:1, "substantial" is overstated, since you need to put FP/FN in context to prevalence

  We will add Matthew's Correlation Coefficient (https://doi.org/10.1016/j.patcog.2019.02.023 ) as it is based on all four classes of the confusion matrix. We will introduce this metric briefly in section 2.5.
  We agree, that "substantial" is too harsh, and will refrain from interpreting the classification at this part of the article.

- L343: not necessarily just resolution, might be bias in the forcing (too wet, too cold)

  We agree to your statement; however, we argue that the elevation bias is the dominant driver of the temperature bias in ERA5L (see Table 2, r = -0.88). We will modify the sentence accordingly.

- L400: again, this is not only resolution!

  Not only, but also. We will soften this statement.

- L474: overstated. You have overall accuracies between 70-90%, which is in range to the correlations for seasonal snow depth. So I assume also shorter-period analyses would be in the same line of accuracy, so you cannot distinguish here by length of the analysed period (only if you actually performed some analyses with your data for 1-2 week periods and then performed a comparison).

  The accuracies are between 70-90%, however not the combination of precision (WRF: 49%, CCLM: 87%, ERA5L: 43%, AMUNDSEN: 61%) and recall (WRF: 86%, CCLM: 62%, ERA5L: 98%, AMUNDSEN: 91%). This results in the $F_1$-scores of WRF: 0.62, CCLM: 0.72, ERA5L: 0.60, AMUNDSEN: 0.73. As the $F_1$-score ignores true negatives, we will add Matthew's Correlation Coefficient (https://doi.org/10.1016/j.patcog.2019.02.023 ). These coefficients amount to 0.49 (ERA5L), 0.51 (WRF), 0.66 (AMUNDSEN), and 0.67 (CCLM).

  One cannot compare this binary classification task ($24^{th}$, $25^{th}$, $26^{th}$ December with snow depth > 1cm) to the performance of seasonal snow depth. We will perform some analyses over shorter periods. We expect that for the winter season mean snow depth evaluation (as in Figs. 6 & 7), deviations over shorter periods may equal out (e.g. less snow in November and too much snow in February may lead to a "correct" mean winter snow depth). However, for tourism applications, often one-week periods are of major interest. For this analysis, we envision error metrics for running 7-day mean snow depths for different elevation categories.

- L488: not new, there have been many SnowMIPs (Essery and co.) showing the same…

  Our study shows this big uncertainty now for a "low-elevation" region, whereas most other studies and SnowMIPs mostly focus on higher elevation. Furthermore, we show the variability in local precipitation and temperature, even if driven by the same large-scale atmospheric conditions (ERA5). We will revise this sentence to be more precise highlighting these two findings.

- L495: So what is better? Use the snow scheme from the climate model? Or take only meteo and apply higher complexity snow models? How does this fit with previous studies that used meteo forcing from climate models to drive snow models?

  There is of course no simple answer. From the modeller's perspective, a coupled snow model (ergo snow scheme from a climate model) would be "better", as the snow simulation feeds back into the climate. However, if the climate forcing from the climate model is biased, the according modelled snow will also be biased. We would argue that high-resolution climate models enlarge the portfolio of tools for impact-relevant research. Analysis of the snow simulations is needed to decide if the climate model snow output might be sufficient, or the whole chain of bias adjustment, statistical downscaling and separate snow model is necessary. We will add that to the conclusion.

- General: Results have a lot of repetition on plots with maps and obs-sim scatter plots. You might consider aggregating the information to prove your point. For example, spatially averaged time series, summary by different elevation, etc.

Melt and accumulation will be removed. The remaining scatterplots will be rearranged with elevation bins. The "white Christmas" section will be enhanced by the new short-period analysis, where the map (Fig. 10) will be moved to the supplement. Instead a summary figure for the model performances for 7-day running windows (as proxy for timeframes relevant for the tourism sector) will be added.

The snow cover duration will be enhanced by the additional validation with MODIS data: MODIS TERRA snow cover (MOD10C1: https://modis-snow-ice.gsfc.nasa.gov/?c=MOD10C1 ) at daily resolution. This evaluation will cover the winter seasons 2000/2001 to 2017/2018. These results will replace / modify Fig. 8.

Hence, repetition on plots with maps and obs-sim scatter plots will decrease.

---

## Author Response (AR1)

**List of relevant changes:**

- Added MODIS snow cover data (2000-2018) for a spatially more representative evaluation
- Added the distinction into two bins (mean snow depth above/below 5cm) for the scatter plots
- Removed snow melt and accumulation due to both reviewer's comments
- Added analysis of the classification for short-duration periods
- Clarification of research aims, choice of study area and setup in the introduction
- Revision of most figures according to both reviewers' suggestions
- Additional discussion of uncertainties (point-vs-grid & snow density)

**Point-by-point answers #1**

Dear Reviewer #1,

We thank you for your constructive critique and the valuable comments and suggestions to improve the quality of the manuscript. We address your comments (dark grey) with our responses (blue) in the following. We think that the outlined additional analysis based on your recommendations will help to improve the study.

Poschlod and Daoz analyze snow depth from two high-resolution climate models and one reanalysis for an area in Southern Germany. The purpose of the study is not clear, since no research aims are stated. The title hints to "suitable for extremes and impact-related research?", however, most of the manuscript is just model evaluation and little on extremes. Some of the impact-related variables are highly questionable. There are some major concerns on parts of the methodology, and the manuscript needs a clearer structure before of publication quality.

**Major points**

- I miss a description of why the research setup. What are your aims and/or hypotheses? Why snow depth? It is not a state variable, and you never discuss how different density estimates might impact your results. Why in-situ observation and not remote sensing? Again, the mismatch of point-vs-grid could be highlighted more clearly, and also the impact of resolution, since I guess a point is more representative for the 1.5km cell than for the 9km cell.

Hypothesis / Aims:

Thank you for this comment. We will better clarify the hypothesis and aims. The main question of the study is: Can new-generation high-resolution regional climate models represent snow depth dynamics at high temporal (daily) and high spatial detail? One according hypothesis is: The high-resolution *dynamical* downscaling of ERA5 atmosphere (via CCLM & WRF) can add value compared to the state-of-the-art ERA5L reanalysis product. This aim is indeed related to "just model evaluation", however at high spatio-temporal scales including extreme snow depth. This is very relevant for impact research.

Further, we try to show the complexity of this topic by highlighting elevation bias, climatic driver biases, albedo, and land cover. We also touched upon the point-vs-grid mismatch (L405-409 old manuscript), but further enhanced the discussion in that regard (L446-L451). We also discussed snow density as uncertainty source (thank you for this hint; see L527-540).

Motivation / Relevance:

Impact research needs information about impactful events at the local scale. Climate change affects the dynamics and conditions, which is why observation-based analyses are limited. Often, coarse-resolution RCMs or even GCMs have been used to drive snow models at local/regional scale. However, bias adjustment, statistical downscaling and the de-coupling of

the interactions of snow dynamics and climate (snow simulations do not feed back into the climate simulation) induce additional uncertainties and limitations.

The "new generation" of high-resolution RCMs could potentially directly provide snow depth information from their internal land surface / snow modules, which leads us to the guiding question: How good are they at representing snow depth?

Setup:

To answer the question and test the hypothesis, we need to:

1) explore high-resolution RCM simulations,
2) which cover not only single years but climatological periods (~ 30 years) to represent the variability and extremes
3) and which are driven by reanalysis in order to be able to compare to observations
4) define a baseline (in our case ERA5L)
5) define a reference (in our case in-situ observations)

1) – 3) strongly limits the choice of available simulations. The CCLM and WRF simulations are the simulations, which we found publicly available.

4) ERA5L as global land reanalysis is the state-of-the-art reanalysis product at 9km resolution, which is also driven by the same climate (ERA5 atmosphere) and therefore comparable.

5) For snow depth, in-situ observations are the typical validation reference (see e.g. https://tc.copernicus.org/articles/15/1343/2021/ ). However, based on your and the other reviewer's suggestion, we add remote sensing data from MODIS Terra (2000-2018) for snow cover validation of the gridded simulations.

In addition to 1) – 5), we added the AMUNDSEN simulations driven by CCLM at the point scale. This setup was added in the course of the evaluation, where the CCLM showed strong systematic underestimation of any snow variable, while representing the climate better than the other models (lowest biases and errors). Hence, driven by the perspective of impact research, we wanted to explore how the separate snow model AMUNDSEN can make use of the well representative CCLM climate.

We extended the motivation for the study, the relevance for the area, and the choice of the setup (L95 – 118).

- L210: In case of heavy snowfall, compaction can be of larger magnitude than melt. (Also how do you derive this index in case of missing observation data - you allowed 30%, right?) Then for maximum accumulation, snowfall might be a better variable than increment in snow depth. And for maximum melt, SWE change (which is anyway the state variable in the model). I suggest removing these analyses. If you want to focus on extreme accumulation and melt, which of course are variables with significant societal impact, then you need to choose appropriate variables. And if you want to use snow depth as proxy instead, at least you have to prove it is meaningful

compared to snowfall and SWE loss. Currently, they cannot be trusted, and therefore should be removed from the manuscript.

We follow your suggestion and remove these analyses from the manuscript. In turn we added Figure 13 (L420) for the evaluation of inter-station extreme snow depth from the supplement.

- Your study focuses on low-elevation snow cover, and your station coverage is rather limited. Especially, all stations are below 1000m. This is important to acknowledge in discussion and research setup.

We emphasize the elevation range of the stations in the Abstract, Introduction and Discussion now. Also, the station coverage is mentioned as uncertainty source (L452-455).

- Consequently, your evaluations, as they are know, are heavily influenced by low snow amounts. For example, you can have high values of relative errors for irrelevant snow amounts. (this is less an issue for high-alpine sites, where you have large snow amounts, related L510). This needs to be discussed. Even better would be analyses that distinguish by elevation or snow amount, or summary metrics/table that take this into account.

Thank you for this suggestion. We divide the evaluation of mean snow depth and annual maxima of snow depth (Figs. 8, 10, 12) into two bins of mean annual snow depths, below/above 5cm. These two bins are also shown in all maps. This distinction shows a clearer picture for e.g. the performance of ERA5L at representing extreme snow depths.

- The discussion on snow albedo as driver of biases is good. However, the important figure is in the supplement and one that is of minor use in the main part.

We add Fig. S9 to the main manuscript as Fig. 16.

**Minor points:**

- L20: improves relative to what? Numbers are somewhat in between.

  Relative to CCLM, which is the driving climate for AMUNDSEN. We revised the abstract.

- L23: "All models fail…" but observations do?

  In the case of white Christmas (snow depth > 1 cm), we would assume that observations can diagnose these conditions. The term "predict" might be misleading in this sentence. We revised the abstract and removed this sentence.

- L25: what limitations?

  The still remaining deviations in intensity and seasonality. We revised the abstract.

  L26: Winter climate is more than just snow.

This statement was supposed to relate to precipitation and temperature and not snow. We clarified that in the revised abstract.

- L26ff: Not sure. Abstract numbers suggest high accuracy, in fact. (after reading the manuscript the low biases make sense, because you only look at low elevations) For climate change research, another important factor is the boundary forcing from GCMs.

  We emphasize the low elevation already in the abstract now and provide metrics later in the article.

- L66: Please be more specific. Blowing snow has already been implemented offline (10.5194/tc-15-743-2021) and also online (10.5194/gmd-16-719-2023).

  Thank you for these references. We were happy to add them.

- L79: Less means there is something, what?

  That was worded a too imprecisely: We haven't found any literature on snow depth extremes in climate models.

- Intro: Research aims are missing.

  see Major Comments; added.

- Intro: Also, I would have expected something on snow studies in climate models, such as 10.3390/atmos10080463, 10.1007/s00382-012-1545-3, 10.5194/tc-17-3617-2023, etc

  Thank you for these recommendations. We added them to the introduction. Especially the extensive third study is very relevant.

- Table 1: would be easier to read if references were put in caption or similar. And ERA5-Land is not statistical downscaling.

  We revised the table layout accordingly.

  The climatic drivers for ERA5-Land result from a linear interpolation (which can be called statistical downscaling). See https://doi.org/10.5194/essd-13-4349-2021: "ERA5-Land is driven by atmospheric forcing derived from ERA5 near-surface meteorology state and flux fields. The meteorological state fields are obtained from the lowest ERA5 model level (level 137), which is 10m above the surface, and include air temperature, specific humidity, wind speed, and surface pressure. The surface fluxes include downward shortwave and longwave radiation and liquid and solid total precipitation. These fields are interpolated from the ERA5 resolution of about 31 km to ERA5-Land resolution of about 9 km via a linear interpolation method based on a triangular mesh."

  Hence, the downscaling of the climate drivers from 31km ERA5 to 9km ERA5L is done via statistical interpolation.

- Sec 2.4: The number of snow stations is quite low, compared e.g. to (10.5194/tc-15-1343-2021). Did you take only stations which had snow, temp, and precip simultaneously? Also the maximum elevation is rather low… Since your study is about snow, maybe it would make more sense to include more observed snow data (not necessarily with temp and precip).

  We first downloaded all stations in the study area via the R package rdwd (as also Matiu et al. 2021 in 10.5194/tc-15-1343-2021), however with the constraint that their measurement extends longer than 1988 (starting winter season of our analysis). That yielded 408 stations. this number seems to be comparable to Matiu et al. However, we remove all stations with more than 30% of missing snow depth data during the period 1987 – 2018. This results in the selection of the subset of stations in the article. Using stations with shorter coverage would distort some of the applied metrics in the article.

- Fig4: I don't see any large dependence between temp and precip (except for ERA5L), so maybe if your point is temp/precip bias depend on elevation biases, it would be better to put elevation bias as continuous x-scale (and not discrete point shape, which makes it hard to read).

  Thank you for this suggestion. We rearranged the figure as proposed.

- FigS2 is quite good and relevant, I suggest moving it into the main manuscript. Just make the elevation bins wider to reduce noise and have a constant line for ERA5L. Also elevation seems not to match between DWD and AMUNDSEN.

  We include Fig. S2 in the main article as Fig. 7 and revised it according to your suggestion. The elevations of DWD and AMUNDSEN do not match as the AMUNDSEN elevation stems from CCLM, as the CCLM simulation is used as climatic driver.

- Plots with obs vs. sim would benefit from a 1:1 line

  Is added.

- L315: since your prevalence is 3:1, "substantial" is overstated, since you need to put FP/FN in context to prevalence

  We add Matthew's Correlation Coefficient (https://doi.org/10.1016/j.patcog.2019.02.023 ) as it is based on all four classes of the confusion matrix. We introduce this metric briefly in section 2.5.
  We agree, that "substantial" is way too harsh, and we refrain from interpreting the classification at this part of the article.

- L343: not necessarily just resolution, might be bias in the forcing (too wet, too cold)

  We agree to your statement; however, we argue that the elevation bias is the dominant driver of the temperature bias in ERA5L (see Table 2, r = -0.88). We modify the sentence accordingly.

- L400: again, this is not only resolution!

Not only, but also. We softened this statement.

- L474: overstated. You have overall accuracies between 70-90%, which is in range to the correlations for seasonal snow depth. So I assume also shorter-period analyses would be in the same line of accuracy, so you cannot distinguish here by length of the analysed period (only if you actually performed some analyses with your data for 1-2 week periods and then performed a comparison).

  The accuracies are between 70-90%, however not the combination of precision (WRF: 49%, CCLM: 87%, ERA5L: 43%, AMUNDSEN: 61%) and recall (WRF: 86%, CCLM: 62%, ERA5L: 98%, AMUNDSEN: 91%). This results in the $F_1$-scores of WRF: 0.62, CCLM: 0.72, ERA5L: 0.60, AMUNDSEN: 0.73. As the $F_1$-score ignores true negatives, we add Matthew's Correlation Coefficient (https://doi.org/10.1016/j.patcog.2019.02.023 ). These coefficients amount to 0.49 (ERA5L), 0.51 (WRF), 0.66 (AMUNDSEN), and 0.67 (CCLM). These numbers prove your point and we removed the too negative interpretation of the classification.

  Still, one cannot compare this binary classification task (24th, 25th, 26th December with snow depth > 1cm) to the performance of seasonal snow depth. We additionally performed some analyses over shorter periods. We added the analysis for the classification of moving 3-day windows with more than 1cm snow depth, as well as a moving 5-day window with more than 10cm of snow depth (Table 4).

  L488: not new, there have been many SnowMIPs (Essery and co.) showing the same…

  Our study shows this big uncertainty now for a "low-elevation" region, whereas most other studies and SnowMIPs mostly focus on higher elevation. Furthermore, we show the variability in local precipitation and temperature, even if driven by the same large-scale atmospheric conditions (ERA5). We revised this sentence to be more precise highlighting these two findings, and not to create the impression that we would want to sell the general finding as "novelty".

- L495: So what is better? Use the snow scheme from the climate model? Or take only meteo and apply higher complexity snow models? How does this fit with previous studies that used meteo forcing from climate models to drive snow models?

  There is of course no simple answer. From the modeller's perspective, a coupled snow model (ergo snow scheme from a climate model) would be "better", as the snow simulation feeds back into the climate. However, if the climate forcing from the climate model is biased, the according modelled snow will also be biased. We would argue that high-resolution climate models enlarge the portfolio of tools for impact-relevant research. Analysis of the snow simulations is needed to decide if the climate model snow output might be sufficient, or the whole chain of bias adjustment, statistical downscaling and separate snow model is necessary.

  In the article, we could at least show, that extreme snow depths are well represented in high-resolution RCMs, which is a novel finding.

General: Results have a lot of repetition on plots with maps and obs-sim scatter plots. You might consider aggregating the information to prove your point. For example, spatially averaged time series, summary by different elevation, etc.

Melt and accumulation is now removed. The remaining scatterplots are rearranged with two mean-snow-depth bins. The "white Christmas" section is enhanced by the new short-period analysis, where the map (Fig. 10) is moved to the supplement (as the spatial pattern is similar to other maps).

The snow cover evaluation is enhanced by the additional validation with MODIS data: MODIS TERRA snow cover (MOD10C1: https://modis-snow-ice.gsfc.nasa.gov/?c=MOD10C1 ) at daily resolution. This evaluation covers the winter seasons 2000/2001 to 2017/2018. These results are added as new Figure 10.

Hence, repetition on plots is decreased.

**Point-by-point answers #2**

Dear Reviewer #2,

We thank you for your valuable comments and suggestions to improve the quality of the manuscript. We address your comments (dark grey) with our responses (blue) in the following. We think that the outlined additional analysis based on your recommendations will help to improve the study.

**General comments**

In this manuscript, variables related to snow cover are compared between station measurements and four models in Southern Germany. The manuscript is mostly well-written with clear figures and presents interesting results. However, some points could improve the paper and strengthen the results.

I think the aims of this study and the justification of the research setup should be more clearly stated.

Aims:

Thank you for this hint. We emphasize the research aims more clearly (L112-118). The main question of the study is: Can new-generation high-resolution regional climate models represent snow depth dynamics at high temporal (daily) and high spatial detail?

How is that motivated and why is this important?

Impact research needs information about impactful events at the local scale. Climate change affects the dynamics and conditions, which is why observation-based analyses are limited. Often, coarse-resolution RCMs or even GCMs have been used to drive snow models at local/regional scale. However, bias adjustment, statistical downscaling and the de-coupling of the interactions of snow dynamics and climate (snow simulations do not feed back into the climate simulation) induce additional uncertainties and limitations.

The "new generation" of high-resolution RCMs could potentially directly provide snow depth information from their internal land surface / snow modules, which leads us to the guiding question: How good are they at representing snow depth?

Setup:

To answer the question, we need to:

1) explore high-resolution RCM simulations,
2) which cover not only single years but climatological periods (~ 30 years) to represent the variability and extremes
3) and which are driven by reanalysis in order to be able to compare to observations
4) define a baseline (in our case ERA5L)
5) define a reference (in our case in-situ observations)

1) – 3) strongly limits the choice of available simulations. The CCLM and WRF simulations are the simulations, which we found publicly available.

4) ERA5L as global land reanalysis is the state-of-the-art reanalysis product at 9km resolution, which is also driven by the same climate (ERA5 atmosphere) and therefore comparable as baseline.

5) For snow depth, in-situ observations are the typical validation reference (see e.g. https://tc.copernicus.org/articles/15/1343/2021/ ). However, based on your suggestion (later in the text) and the other reviewer's suggestion, we will add remote sensing data for snow cover validation of the gridded simulations.

In addition to 1) – 5), we added the AMUNDSEN simulations driven by CCLM at the point scale. This setup was added in the course of the evaluation, where the CCLM showed strong systematic underestimation of almost any snow variable, while representing the climate better than the other models (lowest biases and errors). Hence, driven by the perspective of impact research, we wanted to explore how the separate snow model AMUNDSEN can make use of the well representative CCLM climate.

We added the choice of the setup and the relevance for the study area in L95-112.

The authors have decided to use only one variable (snow depth) instead of using e.g. SWE, even though SWE might be a better variable for estimating snow accumulation and snow melt. They state (L211) that snow melt is assumed as the main driving process for snow depth reduction without adding any reference. I think this should be further discussed.

Based on the suggestion of the other reviewer, we removed this analysis of snow accumulation and melt due to the uncertainties of compaction. We agree that SWE would help to support our analysis. However, this variable is not available from the stored simulations (WRF & CCLM).

Also, I think the authors should state more clearly why this study area (Southern Germany) was chosen. It would be interesting to see the analysis cover also mountainous regions.

The study area is exposed to impactful snow depth dynamics, but not as strongly as alpine regions. The choice of the study area is motivated by the impact perspective. Impact and risk are related to exposure and vulnerability. While snow depths in alpine regions will be more extreme, the exposure in terms of affected people is higher in Southern Germany. We added that in L95-105.
However, we fully agree that such an analysis would also be interesting for alpine areas. The other reviewer has given a reference to https://doi.org/10.5194/tc-17-3617-2023, where a 12.5km and a 2.5km simulation is evaluated over the Alps.

Why are only in situ observations used in the comparison and not remote sensing data? I think the authors should clearly state the justification for using only in situ data. As three of the used models show gridded snow cover estimates, using e.g. satellite-based gridded estimates as a comparison would make sense.

For snow depth, in-situ observations are the typical validation reference (see e.g. https://tc.copernicus.org/articles/15/1343/2021/ ). However, based on your suggestion, we added remote sensing data for snow cover validation of the gridded simulations. We

compared the gridded simulations to MODIS TERRA snow cover (MOD10C1: https://modis-snow-ice.gsfc.nasa.gov/?c=MOD10C1 ) at daily resolution. This evaluation covers the winter seasons 2000/2001 to 2017/2018 (see new Fig. 10).

**Specific comments**

Table 1 is a bit confusing and it is hard to see which model is in which line, as the names that are used throughout the manuscript are not clearly listed. I suggest adding a column on the left with the model name (WRF, CCLM, AMUNDSEN, and ERA5). Also, consider organizing the table in the same order as the text (first Regional climate models, second ERA5-Land, and last AMUNDSEN).

Thanks for this hint, we adjusted the table accordingly and added the model names.

Figure 3. I suggest adding the name of the model on top of each plot instead of just mentioning it in the caption. This applies to all figures throughout the manuscript.

We added the model names for all figures, where subplots are categorized by the model.

Figure 3. Consider adding a diagonal line (from lower left to upper right corner) to each plot so it is easier to see whether there are biases in the models. This also applies to all similar figures throughout the manuscript.

Diagonal is added to all scatterplots.

Figure 4. Consider adding a bit darker horizontal and vertical lines at x=0 and y=0.

We enhanced the 0-lines accordingly (darker/thicker).

Figure 5. Please add somewhere (in the text or in this figure) that you use the acronym DWD for the observations.

Is replaced by "OBS" in all relevant figures and explained in the caption.

Figure 10. Is the difference relative or absolute difference?

The differences of white Christmas are absolute differences of the percentages. We clarify this in the caption.

L381. Change 0.15 m to 15 cm. I think it is better to use the same units that are used in the figure.

True, we adjust all units to cm in the text.

I suggest the authors check the grammar carefully. The manuscript is overall very well-written but some mistakes (e.g. incorrect prepositions) exist.

Thank you for the feedback – we checked the grammar in the course of the revision.

---

## Author Response (AR2)

Dear Reviewer,

We thank you again for the second round of reviewing our manuscript. We are glad that our revision has addressed your comments and suggestions to your satisfaction. We address your remaining comments (dark grey) with our responses (blue) in the following.

• Great motivation of research aims. It would be nice, if you could provide the readers with an obvious summary explicitly saying: "The research aims of this study are 1) ... 2) ..."

Thank you for this suggestion. We add this summary at the end of the introduction section.

• Table 1: Sorry to bother again on the same point. From what I understood, ERA5L in fact uses statistical downscaling to interpolate meteo forcing, but then the land surface scheme is run offline without assimiliation in order to derive land variables (snow, runoff, ...). So, if a physical-based land scheme is run, I'm not sure "statistical downscaling" is the appropriate term. Of course it's also not dynamical downscaling. Maybe it deserves its own distinction?

We try to clarify this issue, by naming the column "Downscaling of the climatic input variables". So, it's now clear for the reader that the downscaling refers only to the climatic input and does not question the physical base of the land surface scheme.

• Good idea to use remote sensing. For the next time, please be aware that cloud cover can induce significant biases, and maybe the cloudgapfilled version of MODIS (MOD10A1F) might be more appropriate. Or the product produced within Snow - ESA Climate Change Initiative, which is explicitly intended for climate model evaluations. With complete series you could also have derived snow cover days from remote sensing (easier comparison than snow cover fraction, which is parametrized differently, and also not always available, as you mentioned).

Thank you for these recommendations.